# Dual targeting of salt inducible kinases and CSF1R uncouples bone formation and bone resorption

Cheng-Chia Tang[1], Christian D Castro Andrade[1], Maureen J O'Meara[1], Sung-Hee Yoon[1], Tadatoshi Sato[1], Daniel J Brooks[1,2], Mary L Bouxsein[1,2], Janaina da Silva Martins[1], Jinhua Wang[3], Nathanael S Gray[3], Barbara Misof[4], Paul Roschger[4], Stephane Boulin[4], Klaus Klaushofer[4], Annegreet Velduis-Vlug[5,6], Yosta Vegting[7], Clifford J Rosen[6], Daniel O'Connell[8], Thomas B Sundberg[8], Ramnik J Xavier[8,9], Peter Ung[10†], Avner Schlessinger[10], Henry M Kronenberg[1], Rebecca Berdeaux[11], Marc Foretz[12], Marc N Wein[1,8,13]*

[1]Endocrine Unit, Massachusetts General Hospital, Harvard Medical School, Boston, United States; [2]Center for Advanced Orthopaedic Studies, Department of Orthopedic Surgery, Beth Israel Deaconess Medical Center, Harvard Medical School, Boston, United States; [3]Dana Farber Cancer Institute, Harvard Medical School, Boston, United States; [4]Ludwig Boltzmann Institute of Osteology at Hanusch Hospital of OEGK and AUVA Trauma Centre, Meidling, 1st Medical Department Hanusch Hospital, Vienna, Austria; [5]Center for Bone Quality, Leiden University Medical Center, Leiden, Netherlands; [6]Center for Clinical and Translational Research, Maine Medical Center Research Institute, Scarborough, Canada; [7]Department of Endocrinology and Metabolism, Academic Medical Center, Amsterdam, Netherlands; [8]Broad Institute of MIT and Harvard, Cambridge, United States; [9]Center for Computational and Integrative Biology, Massachusetts General Hospital, Boston, United States; [10]Department of Pharmacological Sciences, Icahn School of Medicine at Mount Sinai, New York, United States; [11]Department of Integrative Biology and Pharmacology, McGovern Medical School at The University of Texas Health Science Center at Houston (UTHealth), Houston, United States; [12]Université de Paris, Institut Cochin, CNRS, Paris, France; [13]Harvard Stem Cell Institute, Cambridge, United States

*For correspondence: MNWEIN@mgh.harvard.edu

Present address: †Genentech, Inc, San Francisco, United States

**Abstract** Bone formation and resorption are typically coupled, such that the efficacy of anabolic osteoporosis treatments may be limited by bone destruction. The multi-kinase inhibitor YKL-05–099 potently inhibits salt inducible kinases (SIKs) and may represent a promising new class of bone anabolic agents. Here, we report that YKL-05–099 increases bone formation in hypogonadal female mice without increasing bone resorption. Postnatal mice with inducible, global deletion of SIK2 and SIK3 show increased bone mass, increased bone formation, and, distinct from the effects of YKL-05–099, increased bone resorption. No cell-intrinsic role of SIKs in osteoclasts was noted. In addition to blocking SIKs, YKL-05–099 also binds and inhibits CSF1R, the receptor for the osteoclastogenic cytokine M-CSF. Modeling reveals that YKL-05–099 binds to SIK2 and CSF1R in a similar manner. Dual targeting of SIK2/3 and CSF1R induces bone formation without concomitantly increasing bone resorption and thereby may overcome limitations of most current anabolic osteoporosis therapies.

## Introduction

Osteoporosis is a major problem in our aging population, with significant health and economic burden associated with fragility fractures (*Harvey et al., 2010*). Bone mass is determined by the balance between bone formation by osteoblasts and bone resorption by osteoclasts (*Zaidi, 2007*). Osteocytes, terminally differentiated cells of the osteoblast lineage buried deep within mineralized bone matrix, sense hormonal and mechanical cues to bone and in turn regulate the activity of cells on bone surfaces (*Dallas et al., 2013*). The majority of current osteoporosis therapeutics act by slowing down bone resorption, a strategy that most often fails to fully reverse the effects of this disease (*Compston et al., 2019*). Currently, bone anabolic treatment strategies are limited; development of orally available small molecules that stimulate bone formation represents a major unmet medical need (*Estell and Rosen, 2021*). Notably, efficacy of parathyroid hormone-based subcutaneous administration of osteoanabolic agents (teriparatide and abaloparatide) may be blunted by concomitant stimulation of bone resorption (*Bilezikian, 2008*). As such, orally available agents that stimulate bone formation without inducing bone resorption represent the 'holy grail' in osteoporosis drug development.

Parathyroid hormone (PTH) signaling in osteocytes stimulates new bone formation by osteoblasts (*Wein, 2018*). Salt inducible kinases (SIKs) are broadly expressed AMPK family serine/threonine kinases (*Sakamoto et al., 2018*) whose activity is regulated by cAMP signaling (*Wein et al., 2018*). In osteocytes, PTH signaling leads to protein kinase A-mediated phosphorylation of SIK2 and SIK3, a signaling event that suppresses cellular SIK activity (*Wein et al., 2016*). Genetic deletion of SIK2 and SIK3 in osteoblasts and osteocytes dramatically increases trabecular bone mass and causes phenotypic and molecular changes in bone similar to those observed with constitutive PTH receptor action (*Nishimori et al., 2019*). PTH signaling inhibits cellular SIK2/3 function; therefore, small molecule SIK inhibitors such as YKL-05–099 (*Sundberg et al., 2016*) mimic many of the actions of PTH, both in vitro and in vivo in initial studies in young, eugonadal mice (*Wein et al., 2016*).

Despite these advances, major unanswered questions remain regarding small molecule SIK inhibitors as potential therapeutic agents for osteoporosis. First, initial in vivo studies with YKL-05–099 showed increased bone formation (via a PTH-like mechanism) and, surprisingly, *reduced* bone resorption. Typically, bone formation and resorption are tightly coupled (*Sims and Martin, 2020*), and both are increased by PTH. Therefore, one goal of the current study is to define the mechanistic basis underlying the 'uncoupling' anti-resorptive effect of this agent. While YKL-05–099 is a potent SIK inhibitor (*Tarumoto et al., 2020*), this compound also targets several other kinases (*Sundberg et al., 2016*), leaving open the possibility that some of its in vivo activities may be SIK-independent. Kinase inhibitor multi-target pharmacology has been exploited therapeutically for cancers whose growth is dependent on multiple activated kinases (*Dar et al., 2012*), yet this strategy has not been widely explored for use of kinase inhibitors in non-oncologic disease indications (*Ferguson and Gray, 2018*). Second, the safety and efficacy of longer-term YKL-05–099 treatment in a disease-relevant preclinical osteoporosis model remains to be determined. Finally, relevant to therapeutic efforts to develop SIK inhibitors for osteoporosis, the phenotypic consequences of post-natal SIK gene ablation are unknown.

Here, we tested YKL-05–099 in female mice rendered hypogonadal by surgical oophorectomy and observed increased trabecular bone mass, increased bone formation, and reduced bone resorption. Despite these beneficial effects, toxicities of hyperglycemia and nephrotoxicity were noted. Inducible, post-natal SIK2/3 gene deletion caused dramatic bone anabolism without hyperglycemia or BUN elevation, indicating that these side effects were due to inhibition of SIK1 or other targets of YKL-05–099. Notably, inducible, global SIK2/3 gene deletion *increased* bone resorption. While YKL-05–099 potently blocked osteoclast differentiation in vitro, deletion of SIK2/3 or SIK1/2/3 showed no obvious effects on differentiation or function of isolated osteoclast precursors. YKL-05–099 also potently inhibited CSF1R, the receptor for the key osteoclastogenic cytokine M-CSF (*Mun et al., 2020*). Modeling revealed that YKL-05–099 prefers a common conformation of both CSF1R and SIK2. Consistent with these results, YKL-05–099 blocked M-CSF action in myeloid cells. Taken together, these findings demonstrate that the dual target specificity of YKL-05–099 allows this multi-kinase inhibitor to uncouple bone formation and bone resorption.

## Results

### YKL-05–099 increases trabecular bone mass in hypogonadal female mice

We previously showed that the SIK inhibitor YKL-05–099 increased bone formation and bone mass in young, eugonadal mice while simultaneously suppressing osteoclastic bone resorption (*Wein et al., 2016*). Based on these findings, we tested the efficacy of this compound in female mice rendered hypogonadal by surgical removal of the ovaries (OVX, *Figure 1A*), a common preclinical model for post-menopausal osteoporosis. In this study, 12-week-old female C57Bl/6J mice were subjected to sham or OVX surgery. 8 weeks later, mice from each surgical group were randomly divided into three treatment groups for 4 weeks total treatment. We performed side-by-side comparison of YKL-05–099 (18 mg/kg) with human PTH 1–34 (100 mcg/kg). As shown in *Figure 1B–E* and *Supplementary file 1*, YKL-05–099 treatment increased trabecular bone mass in the femur and L5 vertebral body of hypogonadal female mice. Compared to once daily PTH (100 mcg/kg) treatment, YKL-05–099 (18 mg/kg) led to comparable gains in trabecular bone mass. In contrast, this dose of PTH increased cortical bone mass and bone strength. The relationship between cortical bone mass and bone strength was preserved in response to YKL-05–099, indicating that this agent does not cause obvious defects in cortical bone quality (*Figure 1F*, *Supplementary file 1*, *Figure 1—figure supplement 1A,B*, *Supplementary file 2*).

### YKL-05–099 uncouples bone formation and bone resorption in OVX mice

Having established that YKL-05–099 increases trabecular bone mass in OVX mice, we next sought to define the underlying cellular mechanisms. First, fasting serum was collected just prior to sacrifice to measure P1NP (a marker of bone formation) and CTX (a marker of bone resorption). Like once daily PTH treatment, YKL-05–099 treatment increased serum P1NP in both surgical groups (*Figure 2A*). As predicted, PTH treatment also increased bone resorption as measured by serum CTX; however, unlike PTH, YKL-05–099 treatment did not lead to statistically-significant increases in serum CTX (*Figure 2B*).

Next, static and dynamic histomorphometry was performed to investigate the effects of PTH and YKL-05–099 on bone cell numbers and activity at the tissue level on trabecular bone surfaces in the metaphysis of the proximal tibia. Like PTH, YKL-05–099 increased osteoblast numbers and activity (*Figure 2C,E,F–H*, *Supplementary file 3*). In contrast, unlike PTH (which, as expected, tended to increase osteoclast numbers, eroded surface, and serum CTX levels), YKL-05–099 treatment reduced both osteoclast numbers and eroded surface in OVX mice (*Figure 2D*, *Supplementary file 3*). Taken together, these findings demonstrate that systemic YKL-05–099 treatment boosts trabecular bone mass and bone formation as PTH does. However, unlike PTH which predictably tends to stimulate both bone formation and bone resorption, YKL-05–099 treatment only increases bone formation.

An intriguing difference between the effects of PTH and YKL-05–099oc curred at the level of osteoid surface (*Supplementary file 3*). As expected, intermittent PTH treatment in OVX mice led to exuberant new bone formation leading to increased accumulation of unmineralized (osteoid) matrix on trabecular surfaces. Although YKL-05–099 increased osteoblast numbers and bone formation rate (as assessed by dual calcein/demeclocycline labeling), osteoid surface was *not* increased by this treatment. This raised the possibility that YKL-05–099 treatment might both accelerate bone matrix deposition by osteoblasts *and* its subsequent mineralization. This observation prompted us to assess bone mineralization density distribution by quantitative backscattered electron imaging (qBEI) (*Roschger et al., 2008*) in order to assess potential effects of YKL-05–099 on bone matrix mineralization. This methodology is best suited to assess mineralization distribution patterns (BMDD) in cortical bone (where YKL-05–099 action was minimal). Only minor BMDD differences between the groups were observed (*Supplementary file 4*, *Figure 1—figure supplement 1C–H*). Notably, the mean calcium concentration between fluorochrome labels given 7 and 2 days prior to sacrifice (Ca$_{Young}$, which was measured for the evaluation of Ca$_{Low}$) did not differ greatly indicating similar mineralization kinetics between the groups based on this technique. Future study is needed to examine potential effects of YKL-05–099 on matrix mineralization in more detail.

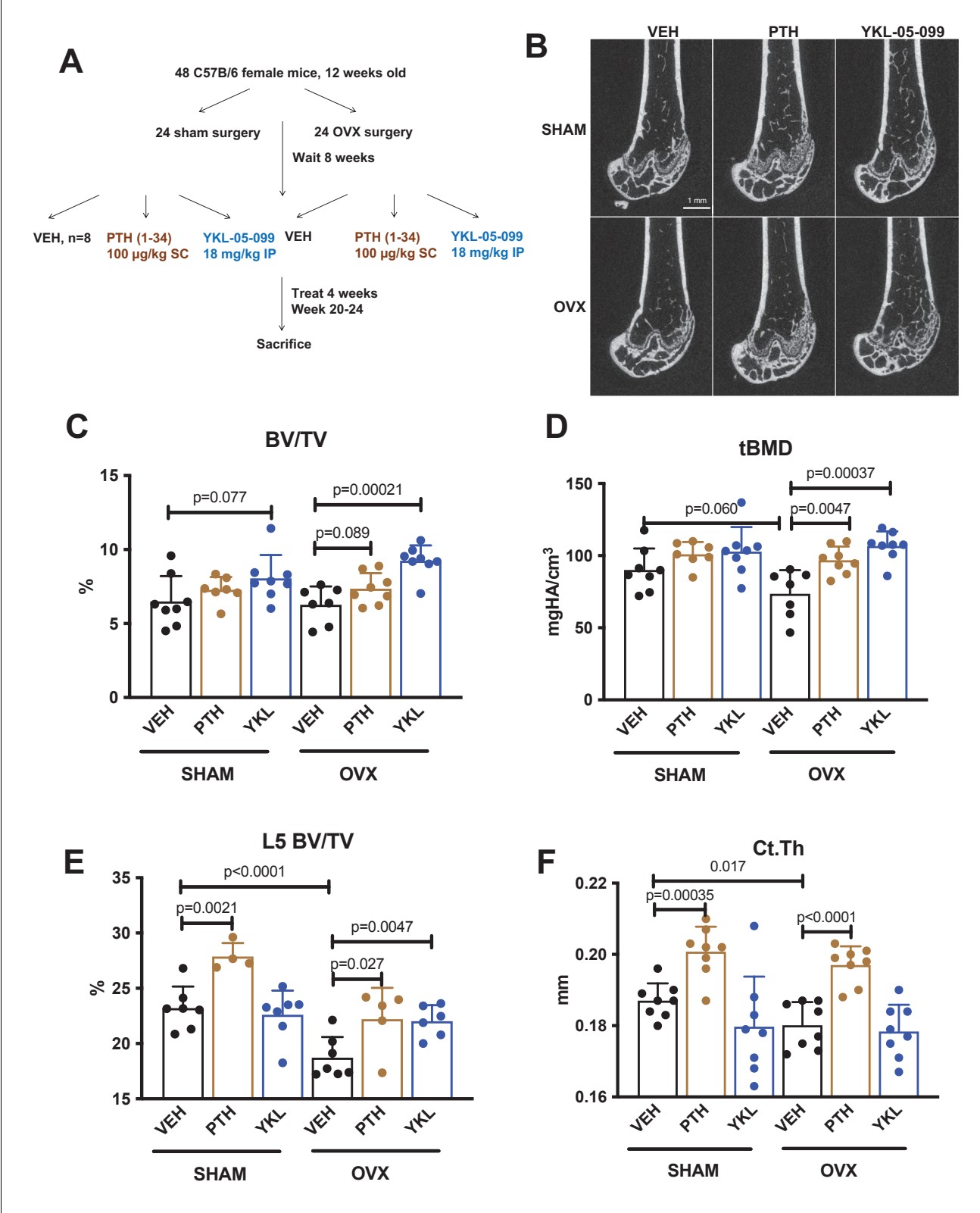

**Figure 1.** YKL-05–099 increases cancellous bone mass in hypogonadal female mice. (A) Overview of ovariectomy (OVX) study design. N = 48 C57B/6 mice were subjected to sham or OVX surgery at 12 weeks of age. Eight weeks later, mice were randomly divided into the six indicated treatment groups, with n = 8 mice per group. Animals were treated over the course of 4 weeks and then sacrificed for skeletal analyses. (B) Representative femur micro-CT images from each treatment group. Scale bar = 1 mm. (C–D) Trabecular parameters in the distal femur. BV/TV = bone vol fraction.

*Figure 1 continued on next page*

Figure 1 continued

tBMD = trabecular bone mineral density. p Values between groups were calculated by two way ANOVA followed by Dunnett's multiple comparisons test. All p values less than 0.1 are shown. OVX surgery reduces trabecular bone mineral density, and this is rescued by YKL-05–099 treatment. (E) Trabecular bone mass in L5. OVX surgery reduces vertebral trabecular bone mass, and this is rescued by YKL-05–099 treatment. (F) Cortical thickness in the femur midshaft. OVX surgery reduces cortical thickness. PTH (100 mcg/kg/d), but not YKL-05–099, increases cortical thickness. Also see *Supplementary file 1* for all micro-CT data from both skeletal sites. All graphs show mean ± SD with each data point representing an individual experimental animal. See Source Data File for additional information.

The online version of this article includes the following figure supplement(s) for figure 1:

**Figure supplement 1.** Results of biomechanical testing (A,B) and quantitative backscattered electron imaging (qBEI, C–H) from OVX study.

A second, provocative effect of YKL-05–099oc curred at the level of bone marrow adipocytes (*de Paula and Rosen, 2020*). PTH signaling in mesenchymal lineage precursors may shift cellular differentiation from adipocyte to osteoblast lineages (*Fan et al., 2017*; *Balani et al., 2017*; *Yang et al., 2019*; *Maridas et al., 2019*). As previously reported (*Yang et al., 2019*), acquired hypogonadism in response to OVX surgery led to increased marrow adipocytes in the metaphyseal region. YKL-05–099 treatment (*Figure 2—figure supplement 1A–C*) reduced marrow adipocyte volume as assessed by semi-automated histology (*Tratwal et al., 2020*). Future studies are needed to define a potential cell intrinsic role for salt inducible kinases (or other intracellular targets of YKL-05–099) in bone marrow adipocyte differentiation and survival.

Given the clear effects observed in trabecular bone in response to YKL-05–099 treatment, we next turned our attention to the safety profile of this agent over the course of 4 weeks of treatment. Surgical treatment group (sham versus OVX) had no impact on parameters measured in vehicle-treated mice; for this reason, data are presented by drug (vehicle, PTH, or YKL-05–099) treatment. YKL-05–099 treatment had no effect on peripheral white blood cell numbers, hemoglobin, platelet counts, or absolute monocyte count (*Figure 2—figure supplement 2A*). Prior to sacrifice, standard toxicology profiling was performed on fasting serum. While most parameters were unaffected by YKL-05–099 treatment, we did note mild but significant increases in blood urea nitrogen (BUN) and glucose (*Figure 2—figure supplement 2B*). Hyperglycemia may be related to a potential role of salt-inducible kinases downstream of hepatic glucagon signaling (*Patel et al., 2014*). In contrast, current genetic models do not necessarily predict nephrotoxicity from in vivo SIK inhibition. Taken together, these results largely demonstrate an appealing therapeutic action in bone in response to YKL-05–099 treatment. However, the potential tolerability issues observed with YKL-05–099 and key differences from the pharmacologic actions of PTH (with respect to bone resorption) prompted us to develop genetic models of adult-onset SIK isoform deletion to gain insight into whether some effects of YKL-05–099 may be related to non-SIK targets of this multi-kinase inhibitor (*Sundberg et al., 2016*).

## Inducible, global SIK2/3 deletion increases trabecular bone mass and increases bone turnover

Similar to YKL-05–099 treatment, deletion of SIK2 and SIK3 in mesenchymal lineage bone cells with Dmp1-Cre increases trabecular bone area and mass (*Nishimori et al., 2019*). However, unlike YKL-05–099 treatment, deletion of SIK2/3 selective in mesenchymal-lineage bone cells dramatically stimulates bone resorption. To mimic the pharmacologic effects of systemic SIK inhibitor treatment, we bred animals with 'floxed' SIK alleles to ubiquitin-Cre$^{ERt2}$ mice (*Ruzankina et al., 2007*) to allow global, tamoxifen-dependent SIK isoform deletion (*Figure 3—figure supplement 1A*). Here, postnatal SIK3 ablation had to be postnatal to circumvent early perinatal lethality due to the key role of this kinase in PTHrP-mediated growth plate hypertrophy (*Nishimori et al., 2019*; *Sasagawa et al., 2012*). For these studies, 6-week-old control (Sik2$^{f/f}$; Sik3$^{f/f}$) and SIK2/3 DKO (Sik2$^{f/f}$; Sik3$^{f/f}$; ubiquitin-Cre$^{ERt2}$) mice were all treated with the same tamoxifen regimen (1 mg IP every other day, three injections total) to control for potential effects of tamoxifen on bone metabolism (*Xie et al., 2020*). Genomic DNA from cortical bone isolated 2 weeks after tamoxifen treatment revealed robust *Sik2* and *Sik3*, but not *Sik1*, deletion (*Figure 3A*). Mice analyzed 3 weeks after tamoxifen treatment showed overt changes in femur morphology including growth plate expansion, increased trabecular bone mass, and increased cortical porosity (*Figure 3B–D*, *Supplementary file 5*). Growth plate

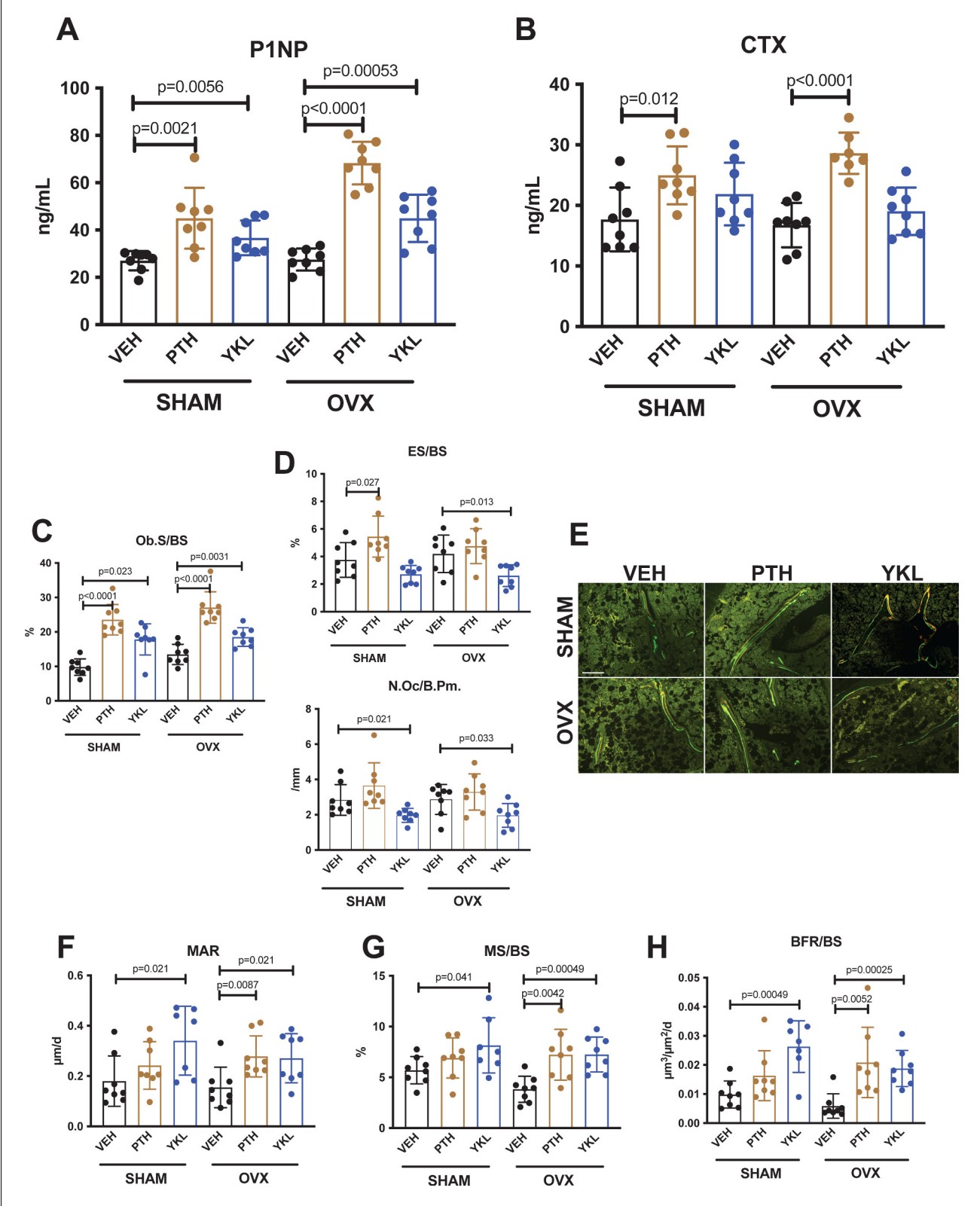

**Figure 2.** YKL-05–099 increases bone formation without increasing bone resorption in OVX mice. (**A, B**) Fasting serum was obtained just prior to sacrifice, following 4 weeks of treatment as indicated. p Values between groups were calculated by two way ANOVA followed by Dunnett's multiple comparisons test. All p values less than 0.05 are shown. Both PTH and YKL-05–099 increase levels of the bone formation marker P1NP. In contrast, only PTH treatment increases levels of the bone resorption marker CTX. P1NP levels (mean ± SD) in the different treatment groups are as follows: SHAM/

*Figure 2 continued on next page*

*Figure 2 continued*

VEH 27.05 (ng/ml)±4.07, SHAM/PTH 44.97 ± 12.9, SHAM/YKL 36.73 ± 7.33, OVX/VEH 27.53 ± 4.64, OVX/PTH 68.29 ± 9.0, OVX/YKL 44.97 ± 10.01. CTX levels (mean ± SD) in the different treatment groups are as follows: SHAM/VEH 17.69 (ng/ml)±5.26, SHAM/PTH 24.97 ± 4.78, SHAM/YKL 21.88 ± 5.18, OVX/VEH 16.75 ± 3.67, OVX/PTH 28.61 ± 3.4, OVX/YKL 19.04 ± 3.9. (C, D) Static histomorphometry was performed on the tibia in the proximal metaphysis to measure cancellous osteoblast surface (Ob.S/BS), eroded surface (ES/BS), and osteoclast number per bone perimeter (Oc.N/B.Pm). Both PTH and YKL-05–099 treatment increases osteoblast surfaces, only PTH tends to increase osteoclast numbers and eroded surface while YKL-05–099 shows the opposite effect. (E) Representative fluorescent images showing dual calcein (green) and demeclocycline (red) labeling on trabecular surfaces. (F–H) Quantification of dynamic histomorphometry parameters: MAR = matrix apposition rate. MS/BS = mineralizing surface per total bone surface. BFR/BS = bone formation rate. Also see *Supplementary file 2* for all histomorphometry data. See Source Data File for additional information. The online version of this article includes the following figure supplement(s) for figure 2:

**Figure supplement 1.** Effects of YKL-05–099 on marrow adipocytes in the proximal tibia.

**Figure supplement 2.** Effects of YKL-05–099 treatment on basic hematologic and serum parameters.

histology (*Figure 3E* and *Figure 3—figure supplement 2*) revealed expansion of proliferating chondrocytes and delayed hypertrophy, an expected phenotype in young, rapidly-growing mice.

Histology and histomorphometry from control and SIK2/3 DKO mice confirmed a dramatic increase in trabecular bone mass and an accumulation of marrow stromal cells (*Figure 4A*, all findings consistent with previously reported phenotypes seen when *Sik2* and *Sik3* are deleted using the *Dmp1*-Cre transgene *Nishimori et al., 2019*). Serum bone turnover markers (P1NP and CTX) showed increased bone formation and increased bone resorption in postnatal-onset, global SIK2/3 DKO animals (*Figure 4B,C*). Consistent with serum markers and histology, histomorphometry and TRAP (tartrate resistance acid phosphatase, osteoclast marker) staining revealed increased osteoblasts, increased osteoclasts, increased bone formation, and increased marrow stromal cell volume (Fb.V/TV) in the SIK2/3 DKOs.

Prompted by the observations that YKL-05–099 treatment caused mild hyperglycemia and increased BUN (*Figure 2—figure supplement 2*), similar serum toxicology profiling was performed in control and postnatal-onset ubiquitous SIK2/3 DKO mice. Animals were treated with tamoxifen at 6 weeks of age and then serum profiling was performed 2 weeks later. In these studies, BUN and fasting glucose levels were unaffected by global SIK2/3 ablation (*Figure 3—figure supplement 1B*). Therefore, these YKL-05–099-associated toxicities are likely due to either SIK1 inhibition or off-target effects of this pharmacologic agent. These reassuring results suggest a favorable initial safety profile associated with whole body SIK2/3 gene deletion, and further support the idea of this target combination for osteoporosis drug development.

Since 6-week-old mice are rapidly growing, we performed additional experiments with postnatal *Sik2/3* gene deletion in older (12-week-old) mice in order to facilitate better comparison between our OVX pharmacologic studies. Notably, similar changes were observed 4 weeks after *Sik2/3* gene deletion in 12-week-old mice including increased trabecular bone (*Figure 3—figure supplement 3A–D*), increased cortical porosity (*Figure 3—figure supplement 3E*), increased levels of bone turnover markers (*Figure 3—figure supplement 3F–H*), and increased marrow stromal cells with abnormal growth plate appearance (*Figure 3—figure supplement 3I*). Similar to studies in 6-week-old animals, *Sik2/3* gene deletion in 12-week-old mice did not affect serum glucose (control 118 ± 24 mg/dL, mutant 151 ± 41 mg/dl, p=0.23) or BUN (control 19.93 ± 4.8 mg/dL, mutant 25.84 ± 3.7 mg/dL, p=0.14) levels.

## No cell-autonomous effects of SIK gene deletion on osteoclast differentiation and function

Discordant effects at the level of bone resorption between *Sik2/3* gene deletion and YKL-05–099 treatment prompted us to study osteoclasts in more detail in this model of presumably ubiquitous inducible SIK ablation. First, we assessed ubiquitin-Cre^ERt2 activity in myeloid osteoclast precursors by crossing ubiquitin-Cre^ERt2 mice with tdTomato^LSL reporter animals. Two weeks after in vivo tamoxifen treatment,>95% of bone marrow myeloid lineage cells (as marked by CD11b and LY6C expression) showed tdTomato expression (*Figure 3—figure supplement 4A*), demonstrating that the ubiquitin-Cre^ERt2 transgene is active in these cells. Bone marrow cells from control and SIK2/3 DKO mice treated with tamoxifen in vivo were isolated and subjected to in vitro osteoclast differentiation using recombinant M-CSF and RANKL (*Figure 3—figure supplement 4B*). Compared to cells

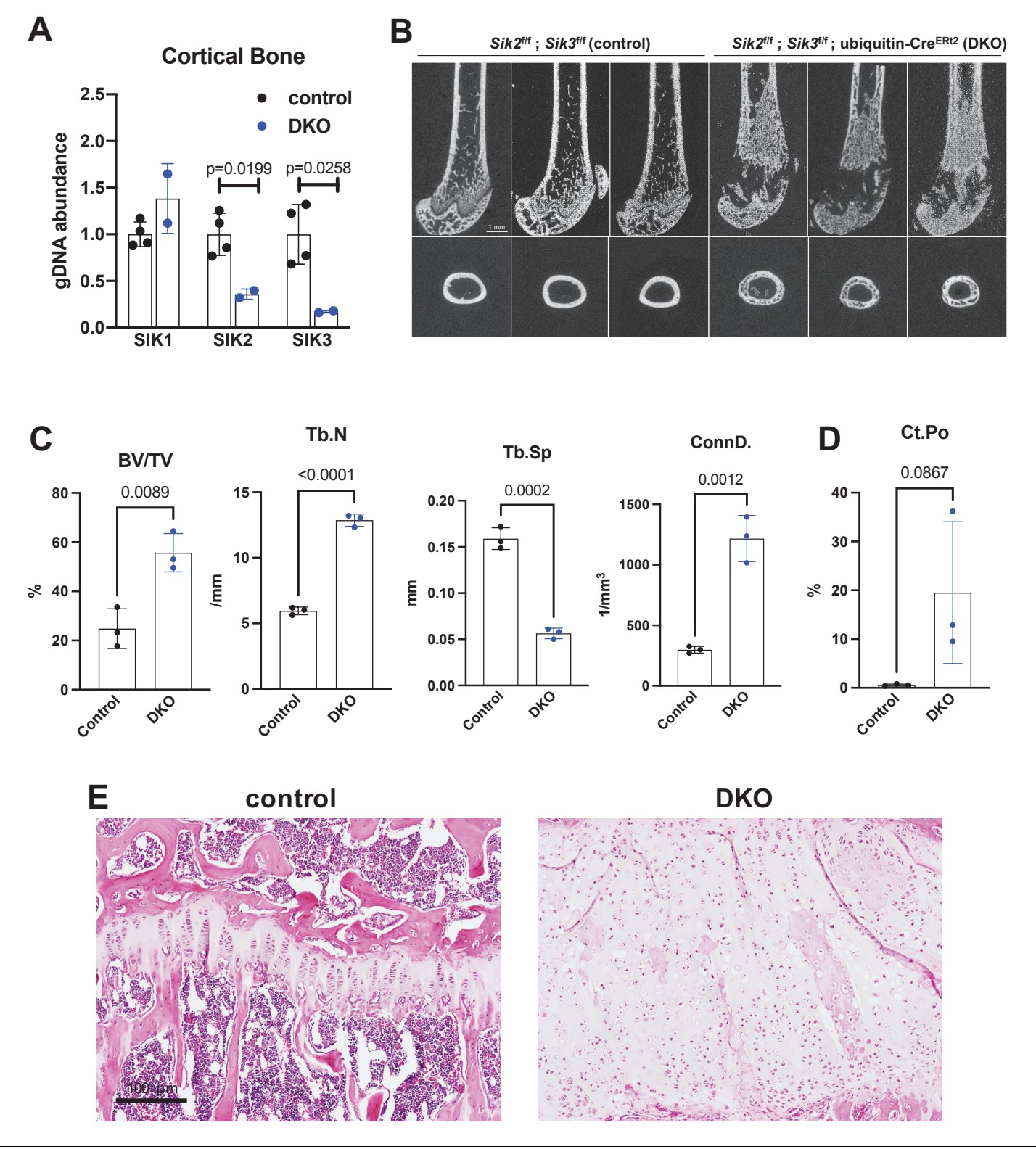

**Figure 3.** Postnatal onset global Sik2/3 deletion increases trabecular bone mass. (**A**) 6 week old *Sik2*^f/f; *Sik3*^f/f (WT) or *Sik2*^f/f; *Sik3*^f/f; ubiquitin-Cre^ERt2 (DKO) mice were treated with tamoxifen (1 mg by intraperitoneal injection, every other day, three doses total) and then sacrificed 14 days after the first tamoxifen injection. Cortical bone genomic DNA was isolated and SIK gene deletion was quantified. (**B–D**) 6 week old *Sik2*^f/f; *Sik3*^f/f (WT) or *Sik2*^f/f; *Sik3*^f/f; ubiquitin-Cre^ERt2 (DKO) mice were treated with tamoxifen (1 mg, IP, Q48H, three doses total) and then sacrificed 21 days after the first tamoxifen

*Figure 3 continued on next page*

*Figure 3 continued*

injection. Micro-CT images (**B**) of the femur show increased trabecular bone mass in the secondary spongiosa, growth plate expansion, and increased cortical porosity. Micro-CT results are quantified in (**C, D**). N = 3 female mice of each genotype were studied. Also see **Supplementary file 5** for all micro-CT parameters measured. (**E**) Tibiae from mice as in (**B**) were stained with hematoxylin and eosin. Representative photomicrographs of the growth plate are shown. Dramatic growth plate expansion and disorganization is observed in inducible Sik2/3 mutant mice, as also demonstrated in **Supplementary file 5**. Scale bar = 100 μm. All graphs show mean ± SD with each data point representing an individual experimental animal. See Source Data File for additional information.

The online version of this article includes the following figure supplement(s) for figure 3:

**Figure supplement 1.** Efficacy and safety of global inducible *Sik2/3* deletion.
**Figure supplement 2.** Growth plate defect induced by post-natal SIK2/3 deletion.
**Figure supplement 3.** Adult-onset global *Sik2/3* deletion increases trabecular bone mass.
**Figure supplement 4.** In vivo *SIK2/3 deletion increases* ex vivo *osteoclast differentiation.*

isolated from control mice, bone marrow cells isolated from SIK2/3 DKO mice showed increased osteoclast differentiation as assessed by increased TRAP secretion and increased numbers of TRAP-positive multinucleated cells (*Figure 3—figure supplement 4C–E*). These results, consistent with evidence of increased osteoclast activity in SIK2/3 DKO mice in vivo yet distinct from what we observed with in vivo YKL-05–099 treatment, led us to further investigate a possible cell-intrinsic role of SIKs in osteoclasts.

Previous studies in murine RAW264.7 pre-osteoclastic cells suggested a potential cell intrinsic role for salt-inducible kinases in osteoclast differentiation (*Lombardi et al., 2017*). Given our apparently discordant observations that (*Harvey et al., 2010*) YKL-05–099 treatment has anti-resorptive effects and (*Zaidi, 2007*) SIK2/3 gene deletion increases bone resorption in vivo, we established a system in which SIK isoforms could be deleted ex vivo to test the cell-intrinsic function of SIKs in osteoclast differentiation and function. For this, bone marrow macrophages from control ($Sik2^{f/f}$; $Sik3^{f/f}$) and SIK2/3 DKO ($Sik2^{f/f}$; $Sik3^{f/f}$; ubiquitin-Cre$^{ERt2}$) mice were treated with 4-hydroxytamoxifen (4-OHT) in vitro (*Chen et al., 2019*) to promote Cre$^{ERt2}$-dependent SIK gene deletion. Using this approach with tdTomato$^{LSL}$; ubiquitin-Cre$^{ERt2}$ bone marrow macrophages, we noted that 4-OHT (0.3 μM, 72 hr) treatment led to tdTomato expression in >95% cells (*Figure 5A*), and robust deletion of *Sik2* and *Sik3* but not *Sik1* (*Figure 5B,C*). Upon M-CSF/RANKL treatment, control and SIK2/3 DKO bone marrow macrophages formed TRAP-positive multinucleated osteoclasts that were able to promote pit resorption on hydroxyapatite-coated surfaces. No significant differences were noted in osteoclast differentiation and function between control and SIK2/3 DKO cells treated with 4-OHT (*Figure 5D–H*).

Since YKL-05–099 inhibits all three SIK isoforms (*Sundberg et al., 2016*), the possibility remained that the inhibitory effects of this compound on osteoclast differentiation were due to SIK1 blockade. Therefore, we crossed $Sik1^{f/f}$ mice (*Nixon et al., 2016*) to $Sik2^{f/f}$; $Sik3^{f/f}$; ubiquitin-Cre$^{ERt2}$ animals to create control and SIK1/2/3 triple knockouts (TKO: $Sik1^{f/f}$; $Sik2^{f/f}$; $Sik3^{f/f}$; ubiquitin-Cre$^{ERt2}$) for generation of bone marrow macrophages. Upon ex vivo 4-OHT treatment, TKO bone marrow macrophages showed >80% SIK gene deletion. Upon M-CSF/RANKL treatment, control and SIK1/2/3 TKO cells formed TRAP-positive multinucleated cells that promoted pit resorption. Similar to SIK2/3 DKO osteoclasts, no significant differences were noted in osteoclast differentiation and function between control and SIK1/2/3 TKO cells (*Figure 6B–F*). Taken together, these data argue against an important cell-intrinsic role for SIKs in osteoclasts. Moreover, these results suggest that increased bone resorption in SIK2/3 DKO mice is due to non-cell autonomous actions of SIK gene deletion, likely due to increased RANKL expression from osteoblast lineage cells similar to effects observed with PTH (*Wein et al., 2016*; *Nishimori et al., 2019*; *Ricarte et al., 2018*). Finally, these genetic data suggest that the anti-resorptive actions of YKL-05–099 may be due to engagement of intracellular target(s) other than salt inducible kinases.

## Modeling reveals conformation-selective preference of YKL-05–099 for SIK2 and CSF1R

Differences between YKL-05–099 treatment and *Sik2/3* gene deletion at the level of bone resorption and blood glucose and BUN prompted us to consider potential 'off target' kinases inhibited by YKL-05–099. First, we used TF-seq (*O'Connell et al., 2016*) to simultaneously profile the action of this

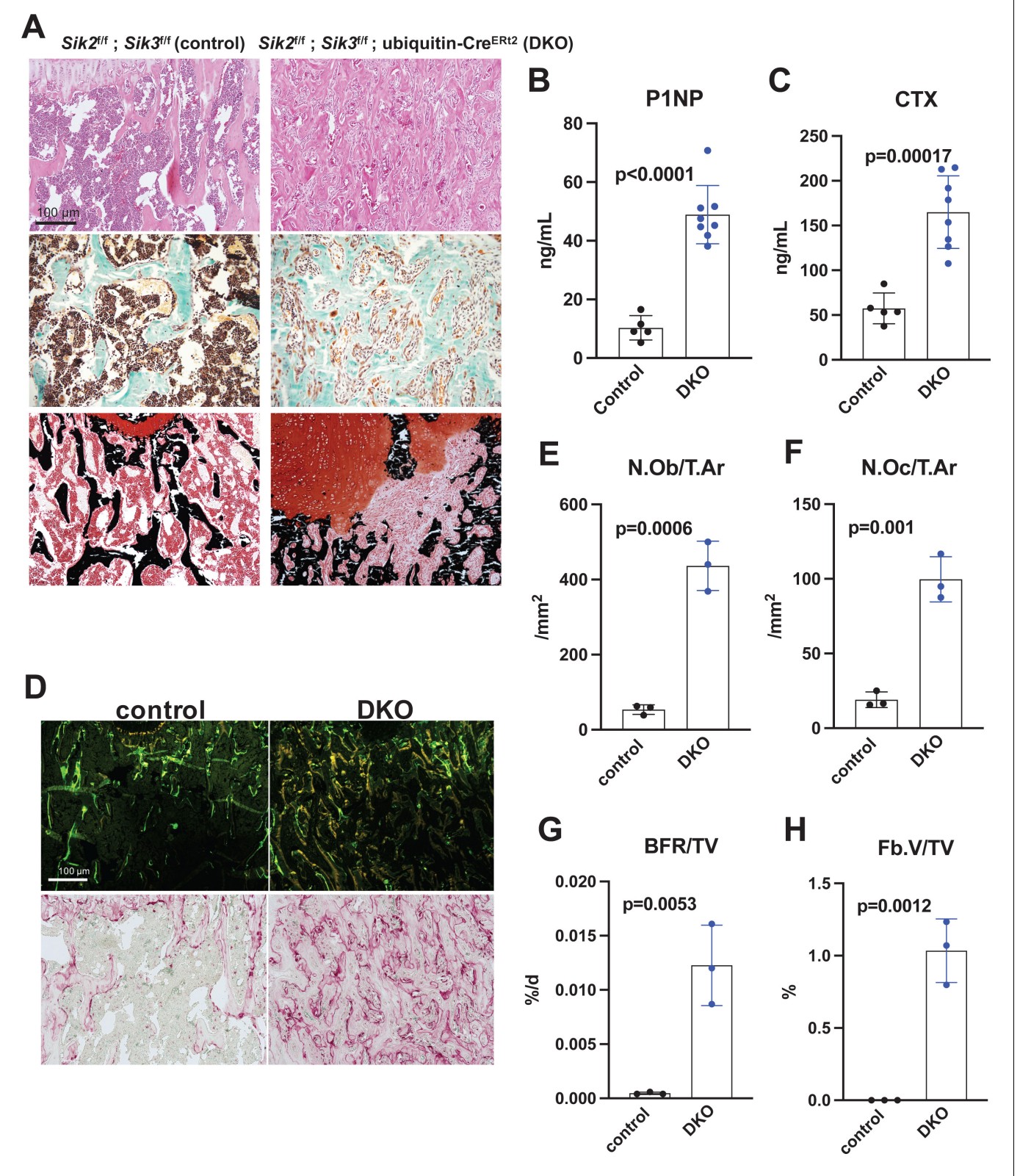

**Figure 4.** Postnatal onset global Sik2/3 deletion increases bone remodeling. (**A**) 6 week old *Sik2*^f/f; *Sik3*^f/f (WT) or *Sik2*^f/f; *Sik3*^f/f; ubiquitin-Cre^ERt2 (DKO) mice were treated with tamoxifen (1 mg by intraperitoneal injection, every other day, three doses total) and then sacrificed 21 days after the first tamoxifen injection. Representative photomicrographs of the proximal tibia are shown. Scale bar = 100 µm. Top panels show hematoxylin and eosin stains on decalcified paraffin-embedded sections revealing increased bone mass and marrow stromal cells in the secondary spongiosa in *Sik2/3* DKO

*Figure 4 continued on next page*

*Figure 4 continued*

mice. Middle panels show trichrome stains demonstrating similar findings on non-decalcified sections used for histomorphometry. Bottom panel shows dual staining with von Kossa and safranin O to demonstrate increased trabecular bone mass, expansion of marrow stromal cells, and growth plate disorganization. Also see *Supplementary file 5* for low power images of growth plate expansion in inducible *Sik2/3* DKO mice. Results shown are representative images from n = 3 female mice per genotype. (B, C) Fasting serum from male mice treated as in (A) were measured for P1NP (bone formation marker) and CTX (bone resorption marker). *Sik2/3* DKO mice show increases in both bone turnover markers. (D) Top panel, mice treated as in (A) were labeled with calcein and demeclocycline at 7 and 2 days prior to sacrifice. Dark field fluorescent images show increased labeling surfaces in *Sik2/3* DKO animals. Bottom panel, decalcified paraffin sections from mice treated as in (A) were stained with TRAP (pink) to label osteoclasts. *Sik2/3* DKO mice show dramatic increases in TRAP + cells present on bone surfaces. (E–H) Quantification of static and dynamic histomorphometry results from mice as in (A). N.Ob/T.Ar=osteoblast number per tissue area. N.Oc/T.Ar=osteoclast number per tissue area. BFR/TV = bone formation rate per tissue volume. Fb.V/TV = fibroplasia vol per tissue volume. See also *Supplementary file 6* for complete histomorphometry data. See Source Data File for additional information.

compound in murine osteocytic Ocy454 cells (*Spatz et al., 2015*; *Wein et al., 2015*) at the level of 58 reporter elements that reflect output from widely-investigated signaling pathways. Using this approach, the only reporter element whose activity was significantly regulated by YKL-05–099 treatment was the CREB-responsive element (*Figure 6—figure supplement 1*). These findings are consistent with a known action of SIKs to regulate the activity of CRTC family CREB coactivators. When phosphorylated by SIKs, CRTC proteins are retained in the cytoplasm. Upon dephosphorylation (in this case in response to SIK inhibitor treatment), CRTC proteins translocate into the nucleus where they potentiate CREB-mediated gene expression (*Sakamoto et al., 2018*; *Wein et al., 2018*; *Altarejos and Montminy, 2011*).

TF-seq profiling of YKL-05–099 failed to inform our thinking about the anti-resorptive effects of this compound in vivo. However, an important clue came from review of previous data profiling YKL-05–099 binding to a panel of 468 recombinant human kinases (*Sundberg et al., 2016*). While this compound binds the active site of SIKs, it also engages a number of tyrosine kinases including the receptor tyrosine kinase CSF1R (FMS, the M-CSF receptor) (*Figure 7A*). Given the essential role of M-CSF action in osteoclast development (*Mun et al., 2020*; *Ross and Teitelbaum, 2005*), this raised the possibility that YKL-05–099 might block osteoclast differentiation via effects on CSF1R.

Most proteins within the human kinome share a highly conserved catalytic domain (*Manning et al., 2002*). This core domain is highly dynamic adopting a range of conformational states that are associated with catalytic activity (*Ung and Schlessinger, 2015*; *Roskoski, 2016*; *Ung et al., 2018*). However, due to the limited number of atomic structures available, most kinases have only been structurally characterized in one or two conformational states (*Ung et al., 2018*). CSF1R (FMS) and SIK2 (QIK) represent two distinct branches of the human kinome (*Figure 7A*): CSF1R belongs to the receptor tyrosine kinase (TK) family, while SIKs are in the CAMK family. Although atomic structures of CSF1R have been solved in two conformational states, structure of SIKs have not been determined. Therefore, to evaluate the putative mode(s) of binding of YKL-05–099, we constructed homology models of CSF1R and SIK2 in the pharmacologically relevant conformational states.

DFGmodel (*Ung and Schlessinger, 2015*) in combination with Kinformation (*Ung et al., 2018*; *Rahman et al., 2019*), can model four commonly observed kinase conformations defined by two critical structural elements of the kinase domain, the αC-helix and the DFG-motif. The four states are αC-in/DFG-in (CIDI), αC-in/DFG-out (CIDO), αC-out/DFG-in (CODI), and αC-out/DFG-out (CODO) conformations. Kinase inhibitors can also be classified by the kinase conformation they bind (*Dar and Shokat, 2011*). For example, type-I inhibitors target the active CIDI conformation, whereas the type-II inhibitor sorafenib and type-I$_{1/2}$ inhibitor erlotinib prefer the inactive CIDO and CODI conformations, respectively (*Ung et al., 2018*). Therefore, to model potential binding modes of a kinase inhibitor, different conformations of the kinase should be explored.

Docking of YKL-05–099 to all four modeled conformations of CSF1R and SIK2 suggested that, across both kinases, this compound shows preferences for the CODI conformation (*Figure 7B*). Therefore, we explored docking modes in the CODI conformation for both kinases. Three common features were noted between how YKL-05–099 engages the CODI conformation of both kinases (*Figure 7C*): (i) the core pyrimidine scaffold interacts with the hinge region of the kinase; (ii) the 2-chloro-6-methylphenyl moiety is buried deep in the binding pocket toward the αC-helix; and (iii) the

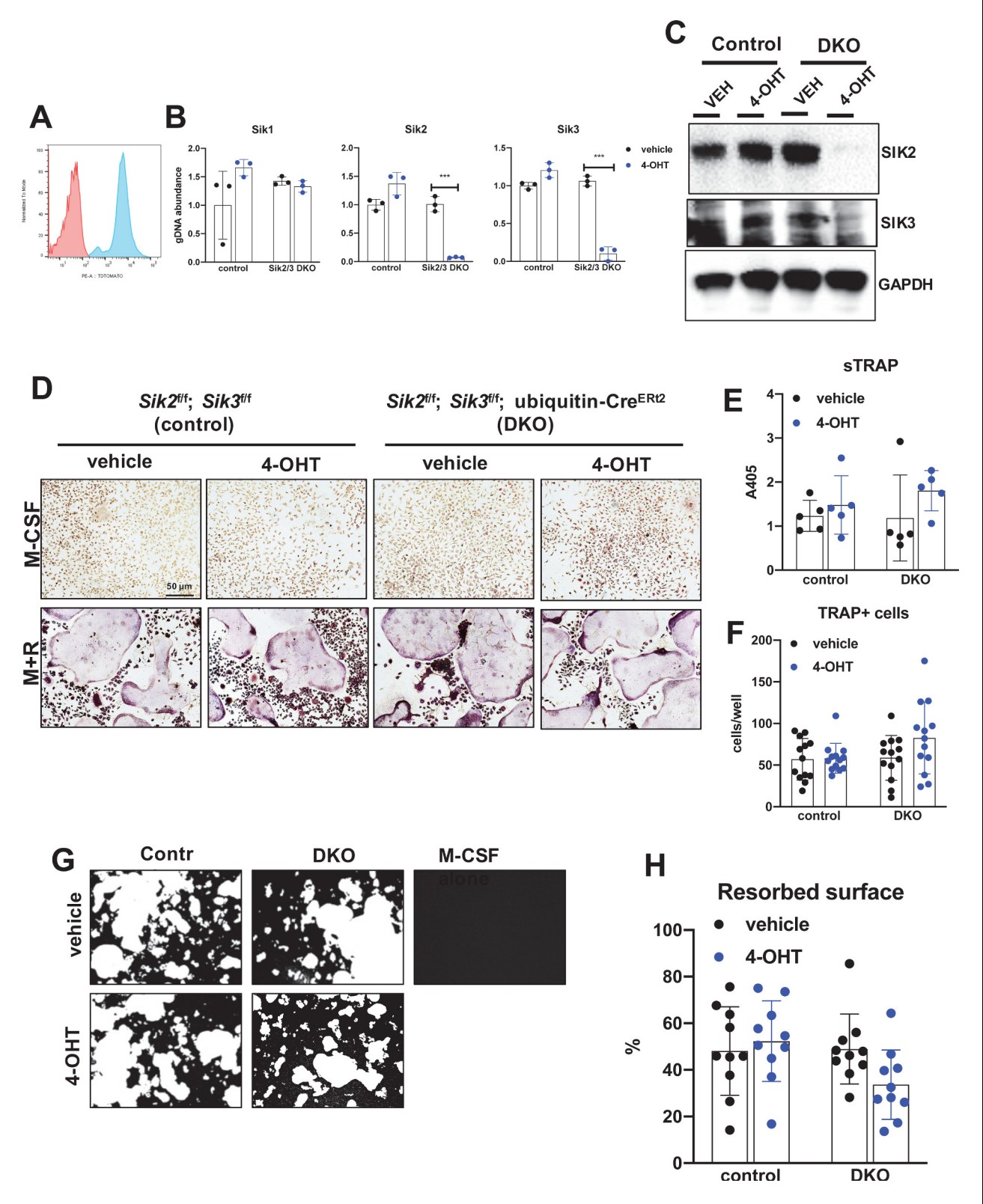

**Figure 5.** Deletion of SIK2 and SIK3 does not affect cell-autonomous osteoclast differentiation or function. (**A**) Bone marrow macrophages from ubiquitin-Cre[ERt2]; Ai14 mice were treated with vehicle (red) or 4-hydroxytamoxifen (blue, 4-OHT, 300 nM) for 72 hr followed by flow cytometry. 4-OHT treatment induces in vitro ubiquitin-Cre[ERt2] activity as measured by this sensitive reporter allele. (**C**) Bone marrow macrophages from control or DKO mice were treated as indicated followed by immunoblotting. 4-OHT treatment leads to robust deletion of SIK2 and SIK3 protein in DKO cells. (**D, E, F**)

*Figure 5 continued on next page*

*Figure 5 continued*

BMMs from control or DKO mice were treated with vehicle or 4-OHT and then subjected to in vitro osteoclast differentiation with M-CSF plus RANKL (M + R). 4-OHT treatment in DKO cells did not cause significant changes in osteoclast differentiation as assessed by morphology (D, scale bar = 50 µm), quantifying TRAP secretion (E), or counting TRAP-positive multi-nucleated cells (F). (G, H) Osteoclasts as in (D) were grown on hydroxyapatite-coated plates in the presence of M-CSF plus RANKL. After 7 days, resorption was measured by von Kossa staining. 4-OHT treatment of DKO cells did not affect resorbed surface. M-CSF treatment alone serves as a negative control to demonstrate that pit resorption in this assay is RANKL-dependent. All in vitro experiments were repeated three times. See Source Data File for additional information.

5-methoxypridin-2-yl moiety resides in the ribose binding site and the 2-methoxy-4-methylpiperdinylphenyl moiety extends outside the binding site. These modeling data demonstrate how a single kinase inhibitor can show promiscuity across unrelated kinases, and provide further support for the model that YKL-05–099 might block bone resorption via effects on CSF1R.

### YKL-05–099 blocks M-CSF action in myeloid cells

Consistent with active site binding data and modeling results, we observed that YKL-05–099 potently blocks CSF1R kinase function in vitro (*Figure 8A*, $IC_{50}$ = 1.17 nM). Bone marrow macrophages were treated with YKL-05–099 during M-CSF/RANKL-stimulated osteoclast differentiation. In these assays, we noted potent inhibition of osteoclast differentiation in response to this compound (*Figure 8B–D*), with cytotoxicity noted at doses above 312 nM (*Figure 8B*). To more directly assess whether YKL-05–099 might block M-CSF action in pre-osteoclasts, we serum-starved cells and then re-challenged with M-CSF. M-CSF-induced receptor Y723 autophosphorylation (*Bourette et al., 1997*) and ERK1/2 phosphorylation were completely blocked by pre-treatment with YKL-05–099 (*Figure 8E*). Consistent with these biochemical effects of YKL-05–099 on M-CSF receptor signaling, we also noted that YKL-05–099 pre-treatment blocked M-CSF-induced upregulation of immediate early genes *Ets2* and *Egr1* in a dose-dependent manner (*Tran et al., 2013*; *Figure 8F,G*, *Figure 8— figure supplement 1*). Finally, we noted that SIK-deficient osteoclasts were equally susceptible to the inhibitory effects of both the potent/selective CSF1R inhibitor PLX-5622 (*Spangenberg et al., 2019*) and YKL-05–099 (*Figure 8H,I*, *Figure 8—figure supplements 2* and *3*). Taken together, these results demonstrate that YKL-05–099 can block M-CSF action in myeloid cells, serving as a likely explanation for the anti-resorptive effect seen with this agent in vivo.

## Discussion

Anti-resorptive therapies (bisphosphonates and the anti-RANKL antibody denosumab) have long represented first line treatment for patients with osteoporosis. These agents boost bone mass and reduce fractures (*Bone et al., 2017*), yet are linked to rare but serious side effects (osteonecrosis of the jaw and atypical femur fractures) which are related to excessive suppression of both bone resorption and bone formation (*Black et al., 2020*). New bone anabolic therapies for osteoporosis are desperately needed in order to provide more effective treatment options for this common and debilitating disease. Currently, analogs of parathyroid hormone (teriparatide and abaloparatide) represent the most commonlyused class of bone anabolic osteoporosis treatment agents. While effective in increasing spine bone density, efficacy of such agents may be limited by concomitant stimulation of bone resorption (*Estell and Rosen, 2021*). For this reason, novel strategies that stimulate bone formation without simultaneous induction of bone resorption are highly desired (*Leder, 2018*). Here, we show that the small molecule kinase inhibitor YKL-05–099 boosts bone formation and trabecular bone mass in a commonly used preclinical model of post-menopausal osteoporosis. In addition to stimulating bone formation, YKL-05–099 treatment inhibits bone resorption. This appealing combination of anabolic and anti-resorptive effects is, to date, only seen with the biologic agent romosozumab (*McClung et al., 2014*), an anti-sclerostin antibody whose widespread use is limited due to risk of increased cardiovascular events (*Fixen and Tunoa, 2021*).

In this study, we investigated mechanisms underlying the in vivo effects of YKL-05–099 treatment, and compared these results with those obtained following post-natal, ubiquitous *Sik2/3* gene deletion. While both organismal perturbations led to increased bone formation and increased trabecular bone mass, key differences were observed. First, YKL-05–099 uncoupled bone formation and bone resorption while *Sik2/3* deletion stimulated both osteoblasts and osteoclasts. High bone resorption

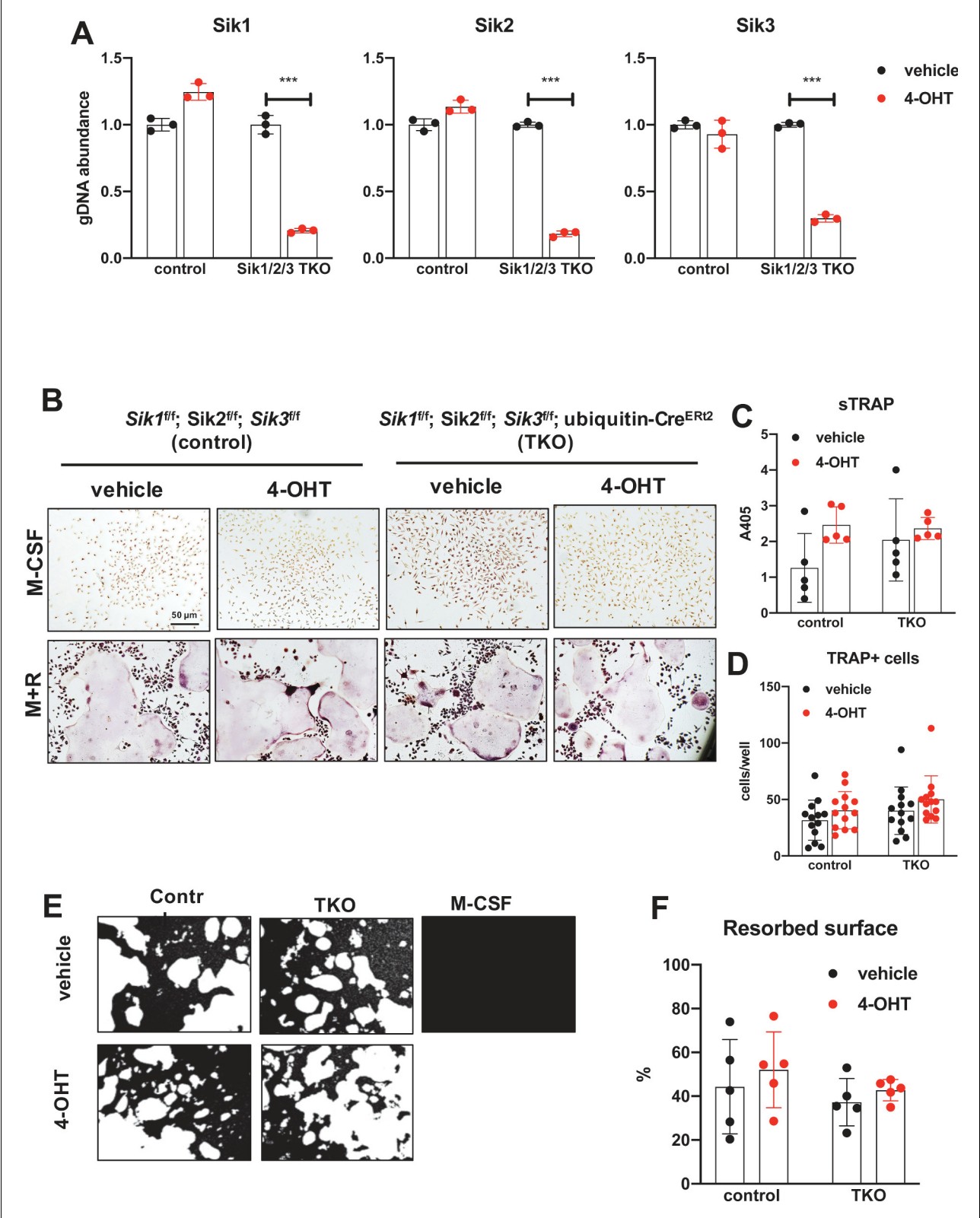

**Figure 6.** Deletion of SIK1, SIK2, and SIK3 does not affect cell-autonomous osteoclast differentiation or function. (**A**) Bone marrow macrophages from ubiquitin-Cre[ERt2]; SIK1/2/3 floxed mice were treated with vehicle (blue) or 4-hydroxytamoxifen (red, 4-OHT, 300 nM) for 72 hr. Genomic DNA was isolated for qPCR-based assessment of SIK isoform deletion. (**B, C, D**) BMMs from control or TKO mice were treated with vehicle or 4-OHT and then subjected to in vitro osteoclast differentiation with M-CSF plus RANKL. 4-OHT treatment in TKO cells did not cause significant changes in osteoclast

*Figure 6 continued on next page*

*Figure 6 continued*

differentiation as assessed by morphology (B), quantifying TRAP secretion (C, scale bar = 50 µm), or counting TRAP-positive multi-nucleated cells (D). (E, F) Osteoclasts as in (B) were grown on hydroxyapatite-coated plates in the presence of M-CSF plus RANKL. After 7 days, resorption was measured by von Kossa staining. 4-OHT treatment of DKO cells did not affect resorbed surface. M-CSF treatment alone serves as a negative control to demonstrate that pit resorption in this assay is RANKL-dependent. All in vitro experiments were repeated three times. See Source Data File for additional information.

The online version of this article includes the following figure supplement(s) for figure 6:

**Figure supplement 1.** Parallel reporter assay (TF-seq) was used to determine the effects of YKL-05–099 on 58 reporter synthetic reporter elements.

seen in response to *Sik2/3* deletion is likely driven by increased RANKL expression by osteoblasts and osteocytes (reviewed in *Wein et al., 2018*). As shown here, we did not observe a cell-intrinsic role for salt-inducible kinases in osteoclast differentiation using ex vivo assays (*Figures 7* and *8*). Rather, YKL-05–099 likely blocks osteoclast differentiation via potent inhibition of CSF1R. As such, our current data support a model in which dual target specificity of YKL-05–099 may explain its ability to uncouple bone formation and resorption in vivo. Interestingly, YKL-05–099 appears to show greater efficacy with respect to increasing trabecular bone mass in OVX mice, suggesting that this compound relies on states of high osteoclast activity for its full bone-building potential.

Second, YKL-05–099 caused mild hyperglycemia and increased BUN, changes not observed following *Sik2/3* deletion. Future studies are needed to better understand the mechanism of these potential (albeit mild) tolerability issues associated with YKL-05–099 treatment. Furthermore, complete characterization of metabolic and renal phenotypes in global/post-natal SIK isoform-selective and compound mutants represents a powerful future approach to better define the physiologic role of these kinases. To date, our studies have focused primarily on the function of SIKs downstream of parathyroid hormone signaling in bone (*Wein et al., 2016*; *Nishimori et al., 2019*). However, potential therapeutic targeting of these kinases for the treatment of cancer, inflammation, and skin pigmentation disorders remains a high priority (*Wein et al., 2018*; *Zhou et al., 2017*). As such, the genetic tools described here to study postnatal roles of SIKs may be valuable reagents across multiple fields, in conjunction with complementary models that address the kinase function of SIK isoforms (*Darling et al., 2017*).

Detailed analysis of bones from OVX mice treated with YKL-05–099 for 4 weeks revealed unanticipated findings including trends toward reduced bone marrow adipocytes and potential effects on matrix mineralization. First, there may be a direct role of PTH signaling in differentiation of bone marrow adipocyte precursors (*Fan et al., 2017*; *Balani et al., 2017*; *Yang et al., 2019*; *Maridas et al., 2019*). In addition, activation of other cAMP-linked GPCR signaling systems, such as ß3-adrenergic receptors, can also inhibit SIK cellular function (*Berggreen et al., 2012*; *Henriksson et al., 2012*) and regulate bone marrow adipocyte size and numbers (*Scheller et al., 2019*). Future studies are needed to determine whether a cell-intrinsic role exists for SIKs in bone marrow adipocyte differentiation or response to catecholamines. Second, we were surprised to note that YKL-05–099 treatment accelerated bone formation (*Figure 2A,H*) without increasing osteoid surface (*Supplementary file 3*). These findings are in stark contrast with the effects of PTH treatment which, as expected, increased bone formation and accumulation of under-mineralized bone matrix. These results suggest that YKL-05–099 somehow accelerates both matrix deposition and decreases osteoid maturation time. The assessment of $Ca_{Young}$ (the mean calcium concentration between the double labels) which represents the calcium level at a well-defined young tissue age, did not show large differences among selected samples from all study groups. However, future studies are needed to investigate how this compound might stimulate mineralization of newly formed bone matrix in more detail (*Murshed and McKee, 2010*). Our previous studies indicated that YKL-05–099 treatment stimulates bone formation via PTH-like effects in osteocytes, including suppressing expression of the osteoblast inhibitor sclerostin (*Wein et al., 2016*). Whether YKL-05–099 has direct effects on osteoblast activity remains to be determined.

Our OVX study demonstrated that YKL-05–099 treatment increased trabecular, but not cortical, bone mass (*Figure 1*). In contrast, sclerostin antibody treatment increases both trabecular and cortical bone mass (*Li et al., 2009*). At this point, we do not understand the mechanistic basis of this compartment-selective effect of this small molecule. Analysis of cortical bone at multiple time points

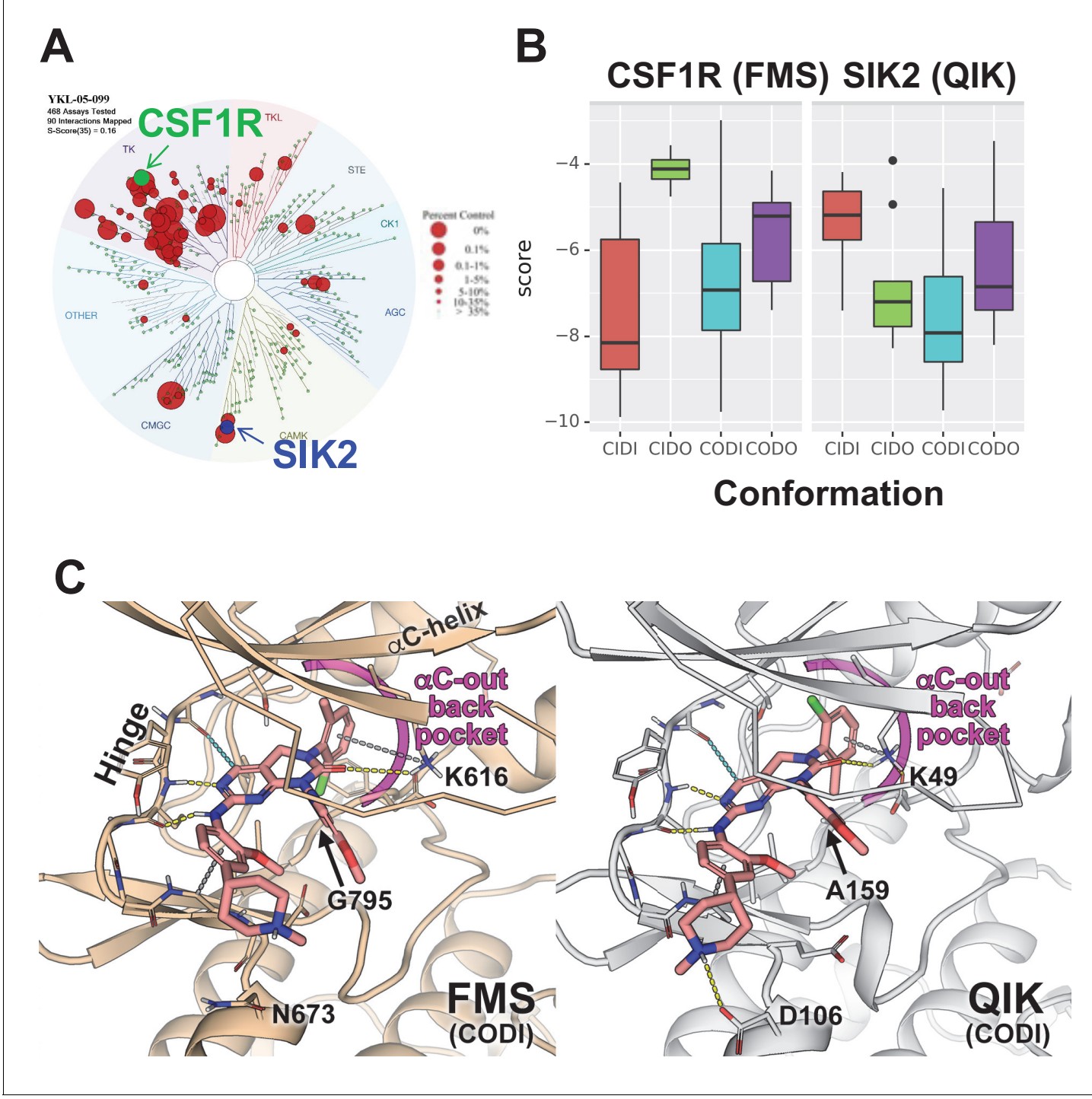

**Figure 7.** Modeling reveals *YKL-05–099* preference for αC-out/DFG-in (CODI) conformation of CSF1R and SIK2. (**A**) Dendrogram representation of previously published (*Sundberg et al., 2016*) kinome profiling data for YKL-05–099 tested at 1.0 µM. Red circles indicate kinases with active site binding to this compound. The position of CSF1R (green) and SIK2 (blue) are noted. (**B**) Docking scores for YKL-05–099 binding to the active site of four kinase conformation defined by the αC-helix and DFG-motif for CSF1R (left) and SIK2 (right). (**C**) Preferential docked pose of YKL-05–099 in the modeled αC-out/DFG-in (CODI) conformation of CSF1R (FMS) (left) and SIK2 (QIK) (right).

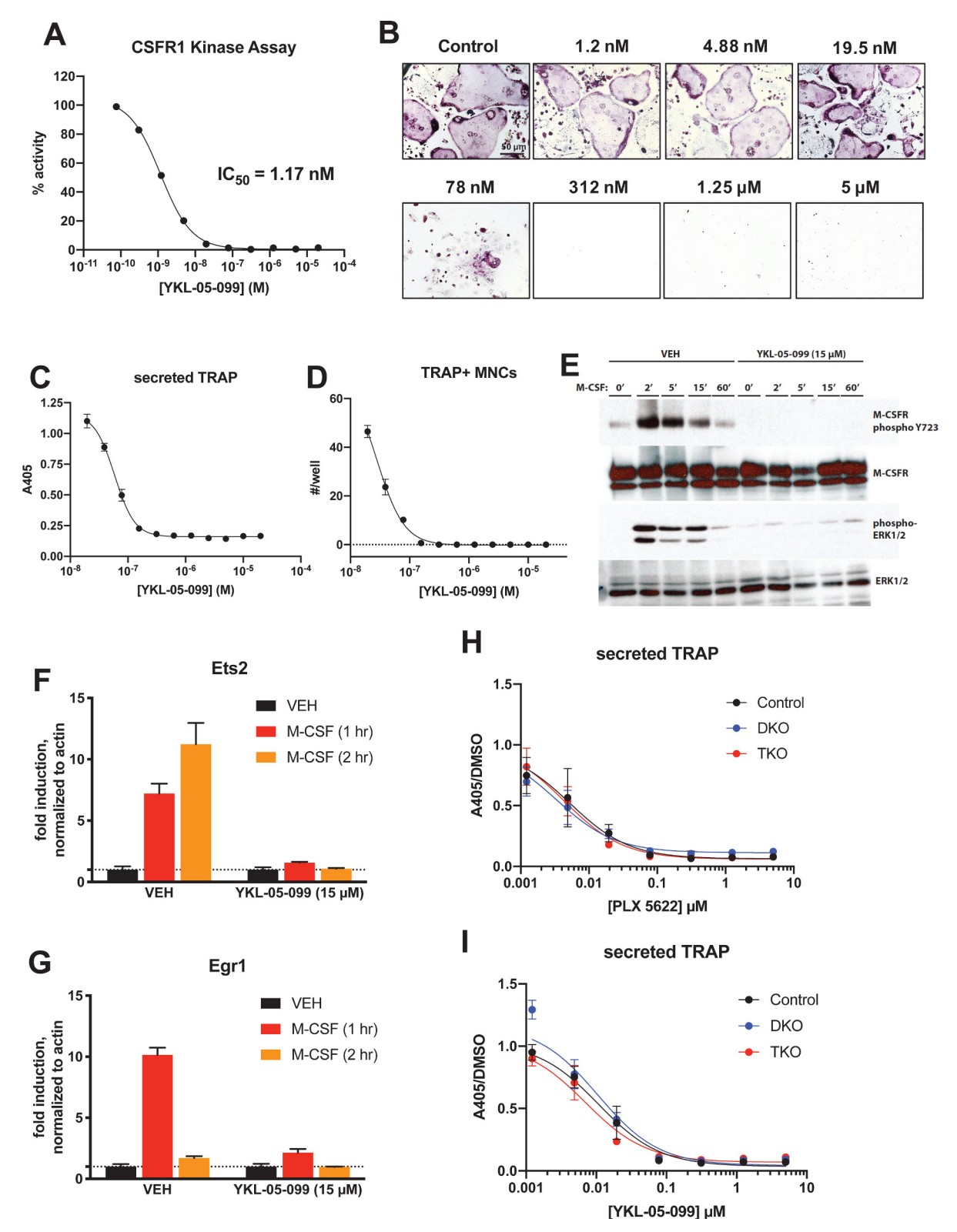

**Figure 8.** YKL-05–099 blocks M-CSF action. (**A**) CSFR1 in vitro kinase assays were performed in the presence of increasing doses of YKL-05–099. This compound blocks CSF1R activity with an $IC_{50}$ of 1.17 nM. (**B**) Murine bone-marrow-derived macrophages were grown in the presence of M-CSF, RANKL, and the indicated doses of YKL-05–099. After 3 days of differentiation, TRAP staining (purple) was performed. YKL-05–099 blocks osteoclast differentiation and causes cytotoxicity in these cultures (scale bar = 50 µm). (**C**) After three days of differentiation in the presence of M-CSF and RANKL,

*Figure 8 continued on next page*

*Figure 8 continued*

conditioned medium was collected and secreted TRAP assays were performed. YKL-05–099 treatment causes a dose-dependent reduction in TRAP secretion. Values indicate mean ± SD of n = 3 wells per condition. (D) After three days of differentiation in the presence of M-CSF and RANKL, TRAP staining was performed. The number of TRAP positive multinucleated cells (MNCs) per well of a 96-well plate (n = 3 wells/condition) is shown. (E) Murine bone marrow macrophages were grown in the presence of M-CSF for 5 days. Cells were then deprived of M-CSF for 6 hr, then pre-treated plus/minus YKL-05–099 (15 µM) for 60 min. Cells were then re-challenged with M-CSF (50 ng/ml) for the indicated times followed by immunoblotting. YKL-05–099 pre-treatment blocks M-CSF-induced M-CSFR autophosphorylation and ERK1/2 phosphorylation. (F, G) Cells as in (E) were challenged with M-CSF (50 ng/ml) for the indicated times, followed by RT-qPCR for the M-CSF target genes Ets2 and Egr1. YKL-05–099 pre-treatment blocks M-CSF-induced Ets2 and Egr1 up-regulation. (H, I) Bone marrow macrophages from ubiquitin-Cre^ERt2; SIK2/3 (DKO) or ubiquitin-Cre^ERt2; SIK1/2/3 (TKO) floxed mice were treated plus/minus 4-OHT as in *Figures 5* and *6*, then plated in M-CSF/RANKL plus the indicated doses of PLX-5622 (H) or YKL-05–099 (I) for 12 days followed by secreted TRAP assays (n = 6 wells from two independent experiments were assayed). In these plots, 'control' cells include BMMs from both DKO and TKO mice treated with vehicle prior to M-CSF/RANKL and inhibitor dose response. Irrespective of the cellular genotype, CSF1R inhibitor (PLX-5622 or YKL-05–099) treatment potently blocked osteoclast differentiation. All in vitro experiments were repeated three times. See Source Data File for additional information.

The online version of this article includes the following figure supplement(s) for figure 8:

**Figure supplement 1.** The dose response relationship of YKL-05–099 on M-CSF action in bone marrow macrophages.
**Figure supplement 2.** PLX-5622 blocks osteoclast differentiation in SIK mutant cells.
**Figure supplement 3.** *YKL-05–099* blocks osteoclast differentiation in SIK mutant cells.

after YKL-05–099 treatment is needed to conclusively demonstrate the absence of a cortical bone effect of this compound. However, *Sik2/3* gene deletion appears to preferentially increase remodeling and bone formation on cancellous bone surfaces (*Figure 4D*). Therefore, it is possible that SIK inhibitors may stimulate remodeling-based bone formation. Further studies using larger animals with dedicated assays to measure modeling versus remodeling-based bone formation (*Ominsky et al., 2014*) are needed to assess this possibility. Moreover, our current studies do not assess whether or not SIK inhibitors, like PTH or sclerostin antibody treatment, stimulate bone formation by activation of previously-quiescent bone lining cells (*Kim et al., 2017*; *Kim et al., 2012*). The relative contribution of sclerostin suppression to YKL-05–099-mediated bone formation also remains to be determined.

Our current claim that YKL-05–099 'uncouples' bone formation and bone resorption is based upon our histomorphometry and serum bone turnover marker data which clearly show that this agent increases bone formation without increasing the measured resorption-related parameters. Future study is needed using dynamic histomorphometry in association with cement line visualization (*Langdahl et al., 2016*) to see if this compound can stimulate modeling-based bone formation independent of bone resorption. While both *Sik2/3* deletion and YKL-05–099 treatment potently stimulate bone formation, we acknowledge that future study is needed to definitively prove that the bone anabolic effect of YKL-05–099 in vivo is due to targeting these two kinases. Currently, reagents (such as mice expressing an inhibitor-resistant SIK2^T96Q allele *Clark et al., 2012*) to rigorously address this question are not available. A final limitation of this OVX study is that our model using C57Bl/6J mice showed obvious OVX-induced loss of trabecular bone volume fraction (BV/TV) in L5 vertebrae but not in the distal femur metaphyseal region (*Figure 1C,E*). Demonstrating OVX-induced loss of trabecular bone in the appendicular skeleton can be challenging, especially in strains (like C57Bl/6J) which show low amounts of trabecular bone at this skeletal site (*Klinck and Boyd, 2008*). Future studies will be needed in other mouse strains and/or robust rat OVX models (*Varela et al., 2017*) to further demonstrate the therapeutic efficacy of SIK inhibitors in bone loss associated with hypogonadism.

Achieving specificity in active site kinase inhibitors is challenging due to the conserved nature of the ATP-binding pocket across protein kinases (*Zhang et al., 2009*). It is plausible that due to small threonine residues (CSF1R: Thr663; SIK2: Thr96) at the so-called gatekeeper position (*Liu et al., 1998*; *Azam et al., 2008*), the active sites of both CSF1R and SIK2 can accommodate multiple inhibitors (*Klaeger et al., 2017*; *Hanson et al., 2019*). Our modeling here further demonstrates that YKL-05–099 preferentially engages the CODI conformation of both kinases in a common binding mode. As such, multi-target binding of kinase inhibitors represents both a challenge and an opportunity as such agents are developed for chronic, non-oncologic indications (*Ferguson and Gray, 2018*). While

avoiding undesired off-target effects will be necessary to avoid unacceptable toxicities, it remains possible that targeting precise combinations of kinases may lead to synergistic therapeutic benefits.

In summary, these findings demonstrate that a single kinase inhibitor can uncouple bone formation and bone resorption via concurrent effects on distinct kinases in distinct cell types. This work provides a framework for development of 'next generation' inhibitors with improved selectivity toward relevant SIK isoforms (likely SIK2 and SIK3) and CSF1R versus the remainder of the kinome. Furthermore, this work highlights the power of combining complementary genetic and pharmacologic approaches to explore the target biology and therapeutic mode of action of a small molecule kinase inhibitor.

# Materials and methods

## Key resources table

| Reagent type (species) or resource | Designation | Source or reference | Identifiers | Additional information |
|---|---|---|---|---|
| Strain, strain background (*Mus musculus* C57BL/6) | *Sik1* floxed | EUCOMM | RRID:MGI_6712189 | Both males and females were used |
| Strain, strain background (*Mus musculus* C57BL/6) | *Sik2* floxed | EUCOMM | RRID:MGI_ 6712188 | Both males and females were used |
| Strain, strain background (*Mus musculus* C57BL/6) | *Sik3*tm1a(EUCOMM)Hmgu | EUCOMM | RRID:MGI_5085429 | Both males and females were used |
| Strain, strain background (*Mus musculus* C57BL/6) | PGK1-FLPo | The Jackson Laboratory | IMSR Cat# JAX:011065, RRID:IMSR_JAX:011065 | Both males and females were used |
| Strain, strain background (*Mus musculus* C57BL/6) | Ubc-CreERt2 | The Jackson Laboratory | IMSR Cat# JAX:008085, RRID:IMSR_JAX:008085 | Both males and females were used |
| Strain, strain background (*Mus musculus* C57BL/6) | Ai14 LSL-tdTomato reporter, Gt(ROSA)26Sor | The Jackson Laboratory | IMSR Cat# JAX:007914, RRID:IMSR_JAX:007914 | Both males and females were used |
| Strain, strain background (*Mus musculus* C57BL/6) | OVX-operated female C57BL/6 | The Jackson Laboratory | IMSR Cat# JAX:000664, RRID:IMSR_JAX:000664 | |
| Antibody | SIK2 (Rabbit monoclonal) | Cell Signaling Technology | Cat# 6919, RRID:AB_10830063 | WB (1:1000) |
| Antibody | Anti-SIK3 (Rabbit polyclonal) | Abcam | Cat# ab88495, RRID:AB_2042747 | WB (1:1000) |
| Antibody | Phospho-M-CSF Receptor Tyr723 (Rabbit monoclonal) | Cell Signaling Technology | Cat# 3155, RRID:AB_2085229 | WB (1:1000) |
| Antibody | CSF1R (Rabbit monoclonal) | Cell Signaling Technology | Cat# 67455, RRID:AB_2799725 | WB (1:1000) |
| Antibody | Phospho-ERK1/2 (Rabbit monoclonal) | Cell Signaling Technology | Cat# 4376, RRID:AB_331772 | WB (1:1000) |
| Antibody | ERK1/2 (Rabbit monoclonal) | Cell Signaling Technology | Cat# 4695, RRID:AB_390779 | WB (1:2000) |
| Antibody | Tubulin (Rabbit monoclonal) | Cell Signaling Technology | Cat# 2128, RRID:AB_823664 | WB (1:1000) |

*Continued on next page*

*Continued*

| Reagent type (species) or resource | Designation | Source or reference | Identifiers | Additional information |
|---|---|---|---|---|
| Antibody | APC anti-mouse/human CD11b antibody | BioLegend | Cat# 101211, RRID:AB_312794 | FC (1:50) |
| Antibody | FITC anti-mouse CD3 antibody | BioLegend | Cat# 100305, RRID:AB_312670 | FC (1:33) |
| Antibody | PE/Cy7 anti-mouse Ly-6C antibody | BioLegend | Cat# 128017, RRID:AB_1732093 | FC (1:75) |
| Antibody | Anti-rabbit IgG, HRP-linked antibody | Cell Signaling Technology | Cat# 7074, RRID:AB_2099233 | WB (1:2500) |
| Commercial assay or kit | pro-collagen type 1 N-terminal peptide (P1NP) | IDS Immunodiagnostic Systems | Cat# AC-33F1, RRID:AB_2801263 | |
| Commercial assay or kit | carboxy-terminal telopeptide of type I collagen (CTX) | IDS Immunodiagnostic Systems | Cat# AC-06F1, RRID:AB_2801265 | |
| Commercial assay or kit | Acid phosphatase leukocyte kit | Sigma-Aldrich | Cat# 387A-1KT | |
| Commercial assay or kit | DNeasy Blood and Tissue Kit | Qiagen | Cat# 69504 | |
| Chemical compound, drug | (Z)—4-Hydroxytamoxifen | Sigma-Aldrich | Cat# H7904-5MG | |
| Chemical compound, drug | Tamoxifen | Sigma-Aldrich | Cat# T5648 | |
| Chemical compound, drug | Methanol BioReagent | Sigma-Aldrich | Cat# 494437–1L | |
| Chemical compound, drug | Dimethyl sulfoxide | Sigma-Aldrich | Cat# D8418-100ML | |
| Chemical compound, drug | Sodium L-tartrate dibasic dihydrate | Sigma-Aldrich | Cat# 228729–100G | |
| Chemical compound, drug | Phosphatase substrate | Sigma-Aldrich | Cat# SRE0026 | |
| Chemical compound, drug | Acetone | VWR | Cat# E6460500mL | |
| Chemical compound, drug | 140 Proof Pure Ethanol | Koptec | Cat# UN1170 | |
| Chemical compound, drug | Formaldehyde 37% by weight with preservative | Fisher Chemical | Cat# F79-500 | |
| Chemical compound, drug | phosphatase inhibitor cocktail | Thermo Fisher Scientific | Cat# 78440 | |
| Chemical compound, drug | PLX5622 | MCE MedChem Express | Cat# HY-114153 | |
| Chemical compound, drug | Calcein | Sigma-Aldrich | Cat# C0875-5G | |

*Continued on next page*

*Continued*

| Reagent type (species) or resource | Designation | Source or reference | Identifiers | Additional information |
|---|---|---|---|---|
| Chemical compound, drug | Demeclocycline hydrochloride | Sigma-Aldrich | Cat# D6140-1G | |
| Peptide, recombinant protein | Recombinant mouse RANKL | R and D systems | Cat# 462-TEC-010 | |
| Peptide, recombinant protein | Recombinant mouse M-CSF | R and D systems | Cat# 416 ML-050 | |
| Peptide, recombinant protein | Recombinant CSFR1 | Invitrogen | Cat# PV3249 | |
| Software, algorithm | HeskaView Integrated software | Heska | | |
| Software, algorithm | FlowJo | BD Biosciences | RRID:SCR_008520 | |
| Software, algorithm | ImageJ | PMID:22930834 | NIH Image, RRID:SCR_003073 | |
| Software, algorithm | Prism 8.0 | GraphPad Software | RRID:SCR_002798 | |
| Other | Bio-Rad Protein Assay Dye Reagent Concentrate | Bio-Rad | Cat# 5000006 | |
| Other | Pierce ECL Plus Western Blotting Substrate | Thermo Scientific | Cat# 32132 | |
| Other | Maxima reverse transcriptase | Thermo Fisher Scientific | Cat# EP0742 | |
| Other | Sunflower oil | Sigma-Aldrich | Cat# 88921–250 ML-F | |
| Other | Red blood cell lysis buffer | Sigma-Aldrich | Cat# R7757-100mL | |
| Other | 0.05% Trypsin-EDTA (1x) | Gibco | Cat# 25300–062 | |

## Mice

All animals were housed in the Center for Comparative Medicine at the Massachusetts General Hospital and all experiments were approved by the hospital's Subcommittee on Research Animal Care. The following published genetically modified strains were used: *Sik1* floxed mice (RRID:MGI: 5648544) (*Nixon et al., 2016*), and *Sik2* floxed mice (RRID:MGI:5905012) (*Patel et al., 2014*). *Sik3*tm1a(EUCOMM)Hmgu mice (RRID:MGI:5085429) were purchased from EUCOMM and bred to PGK1-FLPo mice (JAX #011065) in order to generate mice bearing a loxP-flanked *Sik3* allele (*Nishimori et al., 2019*). Ubiquitin-CreERt2 mice (*Ruzankina et al., 2007* JAX #008085) were intercrossed to *Sik1/2/3* floxed mice. In some instances, ubiquitin-CreERt2 mice were intercrossed to Ai14 tdTomatoLSL reporter mice (JAX #007914). CreERt2-negative littermate controls were used for all studies to account for potential influence of genetic background and impact of tamoxifen on bone homeostasis. All mice were back-crossed to C57Bl/6J for at least five generations.

Both males and females were included in this study, except for the OVX studies where only female mice were used. All procedures involving animals were performed in accordance with guidelines issued by the Institutional Animal Care and Use Committees (IACUC) in the Center for Comparative Medicine at the Massachusetts General Hospital and Harvard Medical School under approved Animal Use Protocols (2019N000201). All animals were housed in the Center for Comparative

Medicine at the Massachusetts General Hospital (21.9 ± 0.8°C, 45 ± 15% humidity, and 12 hr light cycle 7 am–7 pm).

For OVX studies, 12-week-old sham- and OVX-operated female C57Bl/6J mice were obtained from a commercial vendor (JAX #000664). Mice from each surgical group were randomly allocated into three drug treatment groups. Drug treatments started 8 weeks after OVX surgery for a total of 4 weeks. YKL-05–099 was dissolved in PBS + 25 mM HCl and injected IP once daily five times per week for a total of 20 injections. PTH was dissolved in buffer (10 mM citric acid, 150 mM NaCl, 0.05% Tween-80, pH 5.0) and injected SC once daily five times per week for a total of 20 injections. Power calculations were performed based on previous data where eugonadal mice were treated with YKL-05–099 for 2 weeks (*Wein et al., 2016*), detailed below. For experiments in which mice were treated with either vehicle or PTH (or YKL-05–099), mice were assigned to alternating treatment groups in consecutive order. Tamoxifen (Sigma-Aldrich, St. Louis, MO, catalog #T5648) was dissolved in 100% ethanol at a concentration of 10 mg/mL. Thereafter, equal volume of sunflower oil (Sigma, catalog #88921–250 ML-F) was added, the solution was vortexed and placed un-capped in a 65°C incubator overnight in order to evaporate ethanol. This working solution of tamoxifen dissolved in sunflower oil at a concentration of 10 mg/mL was used for intraperitoneal injections.

The sample size for this study was determined using the following power calculation. *Grassi et al., 2016* performed a similar study design to assess the effects of the small molecule $H_2S$ donor GYY4137 on OVX-induced bone loss. In these studies, femoral BV/TV (%± SD) fell from 7.8 ± 2 to 4.2 ± 1 following OVX. From these studies, a sample size of n = 5/group would be required to reach 95% power to detect a 'p' value of 0.05. In our published studies (*Wein et al., 2016*) with YKL-05-099 (6 mg/kg/d) treatment caused BV/TV to increase from 10.6 ± 1.0 to 12.5 ± 1.0. In these studies, a sample size of n = 7/group would be required to reach 95% power to detect a 'p' value of 0.05. In our preliminary data using YKL-05–099 18 mg/kg/d caused BV/TV to increase from 10.3 ± 2.0 to 16.3 ± 2.0. In these studies, a sample size of n = 3/group would be required to reach 95% power to detect a 'p' value of 0.05. In designing these experiments, a conservative estimate will be used of n = 8/group, as this number reflects one additional mouse per group beyond the minimum number needed from the aforementioned scenarios. In addition, since more stringent ANOVA analysis will be required to test for an interaction between surgery and drug treatment, a larger size will be necessary. For studies investigating the effects of *Sik2/3* gene deletion on bone parameters, no statistical methods were used to predetermine sample size. The sample size was determined based on our previous experience characterizing the skeletal effects of *Sik2/3* gene deletion (*Nishimori et al., 2019*).

## Antibodies and compounds

YKL-05–099 was synthesized as previously described (*Sundberg et al., 2016*), PLX-5622 was obtained from MedChem Express (Monmouth Junction, NJ). Antibody sources and dilutions are listed below under the immunoblotting section. See Key Resources Table.

## Micro-CT

Assessment of bone morphology and microarchitecture was performed with high-resolution micro-computed tomography (μCT40; Scanco Medical, Brüttisellen, Switzerland) in 8-week-old male mice. Femora and vertebrae were dissected, fixed overnight in neutral buffered formalin, then stored in 70% EtOH until the time of scanning. In brief, the distal femoral metaphysis and mid-diaphysis were scanned using 70 kVp peak X-ray tube potential, 113 mAs X-ray tube current, 200 ms integration time, and 10 μm isotropic voxel size. Cancellous (trabecular) bone was assessed in the distal metaphysis and cortical bone was assessed in the mid-diaphysis. The femoral metaphysis region began 1700 μm proximal to the distal growth plate and extended 1500 μm distally. Cancellous bone was separated from cortical bone with a semiautomated contouring program. For the cancellous bone region, we assessed trabecular bone volume fraction (Tb.BV/TV, %), trabecular thickness (Tb.Th, mm), trabecular separation (Tb.Sp, mm), trabecular number (Tb.N, 1/mm), connectivity density (Conn.D, $1/mm^3$), and structure model index. Transverse μCT slices were also acquired in a 500 μm long region at the femoral mid-diaphysis to assess total cross-sectional area, cortical bone area, and medullary area (Tt.Ar, Ct.Ar and Ma.Ar, respectively, all $mm^2$); cortical bone area fraction (Ct.Ar/Tt.Ar, %), cortical thickness (Ct.Th, mm), porosity (Ct.Po, %) and minimum ($I_{min}$, $mm^4$), maximum ($I_{max}$,

mm$^4$) and polar ($J$, mm$^4$) moments of inertia. Bone was segmented from soft tissue using fixed thresholds of 300 mg HA/cm$^3$ and 700 mg HA/cm$^3$ for trabecular and cortical bone, respectively. Scanning and analyses adhered to the guidelines for the use of micro-CT for the assessment of bone architecture in rodents (*Bouxsein et al., 2010*). Micro-CT analysis was done in a completely blinded manner with all mice assigned to coded sample numbers. Samples from some mice were unable to be measured due to dissection artifacts. All measured samples were included in subsequent analyses.

## Mechanical testing

Femora were mechanically tested in three-point bending using a materials testing machine (Electroforce 3230, Bose Corporation, Eden Prairie, MN). The bending fixture had a bottom span length of 8 mm. The test was performed in displacement control moving at a rate of 0.03 mm/s with force and displacement data collected at 50 Hz. All bones were positioned in the same orientation during testing with the cranial surface resting on the supports and being loaded in tension. Bending rigidity (EI, N-mm2), apparent modulus of elasticity (E$_{app}$, MPa), and ultimate moment (M$_{ult}$, N-mm) were calculated based on the force and displacement data from the tests and the mid-diaphysis bone geometry measured with µCT. Bending rigidity was calculated using the linear portion of the force-displacement curve. The minimum moment of inertia (I$_{min}$) was used when calculating the apparent modulus of elasticity.

## Histomorphometry

Femora were subjected to bone histomorphometric analysis. The mice were given calcein (20 mg/kg by intraperitoneal injection) and demeclocycline (40 mg/kg by intraperitoneal injection) on 7 and 2 days before necropsy, respectively. The femur was dissected and fixed in 70% ethanol for 3 days. Fixed bones were dehydrated in graded ethanol, then infiltrated and embedded in methylmethacrylate without demineralization. Undecalcified 5 µm and 10-µm-thick longitudinal sections were obtained using a microtome (RM2255, Leica Biosystems., IL, USA). Five µm sections were stained with Goldner Trichome and at least two nonconsecutive sections per sample were examined for measurement of cellular parameters. The 10 µm sections were left unstained for measurement of dynamic parameters, and only double-labels were measured, avoiding nonspecific fluorochrome labeling. A standard dynamic bone histomorphometric analysis of the tibial metaphysis was done using the Osteomeasure analyzing system (Osteometrics Inc, Decatur, GA, USA). Measurements were performed in the area of secondary spongiosa, 200 µm below the proximal growth plate. The observer was blinded to the experimental genotype at the time of measurement. The structural, dynamic, and cellular parameters were calculated and expressed according to the standardized nomenclature (*Dempster et al., 2013*).

For the adipocyte parameters, we used sections of the proximal tibia with H and E staining at ×20 magnification. The following MAT outcome parameters were measured and calculated: [1] MAT volume as a percentage of the tissue volume (total adipose tissue volume: Ad.V/TV; %), [2] MAT volume as a percentage of the marrow volume (marrow adipose tissue volume: Ad.V/Ma.V; %), [3] adipocyte density (Ad.Dn; cells/mm2 marrow area) representing adipocyte number. These measurements were performed by semi-automatically tracing out individual adipocytes 'ghosts' in all the fields analyzed. Adipocyte ghosts appear as distinct, translucent, yellow ellipsoids in the marrow space. The total proximal tibia area below the secondary spongiosa was measured in 1–4 sections per biopsy. Adipocyte analysis was performed using a semi-automated measurement program on ImageJ (*Schneider et al., 2012*) based image analysis software adapted from the OsteoidHisto package (*van 't Hof et al., 2017*). All assessments of the sections were performed together by examiners (YV and AV-V) who were blinded to the intervention assignment.

## Serum analysis

Three hr fasting serum was collected at ZT3 from mice just prior to sacrifice by retro-orbital bleed. Serum was isolated and analyte levels were determined using the following commercially available detection kits: P1NP (IDS Immunodiagnostic Systems, #AC-33F1), CTX from IDS (#AC-06F1). All absolute concentrations were determined based on interpolation from standard curves provided by the manufacturer. DRI-CHEM 700 veterinary chemistry analyzer (Heska, Loveland, CO) was used for

measurement of serum analytes: albumin, alkaline phosphatase, ALT, BUN, calcium, cholesterol, globulin, glucose, phosphorus, total bilirubin, triglycerides, and total protein. For complete blood counts, whole blood was collected into heparin-coated tubes and kept on ice. CBCs were measured on a 2015 Heska Element HT5 Veterinary Hematology Analyzer. HeskaView Integrated software was used for data analysis.

## Quantitative Backscattered Electron Imaging (qBEI)

Distal femora from six study groups were analyzed (n = 8 each group): SHAM VEH, SHAM PTH, SHAM YKL, OVX VEH, OVX PTH, and OVX YKL. These bones were embedded undecalcified in poly-methylmethacrylate (PMMA) and measured for bone mineralization density distribution (BMDD) using qBEI. The surfaces of the sample blocks were flattened by grinding and polishing (Logitech PM5, Glasgow, Scotland) and carbon coated so as to facilitate qBEI. A scanning electron microscope equipped with a four quadrant semiconductor backscatter electron detector (Zeiss Supra 40, Ober-kochen, Germany) was used. Areas of metaphyseal (MS) and cortical midshaft bone (Ct) were imaged with a spatial resolution of 0.88 μm/pixel. The gray levels, reflecting the calcium content, were calibrated by the material contrast of pure Carbon and Aluminum. Thus, the resulting gray level histograms could be transformed into calcium weight percent (wt% Ca) histograms (*Figure 1—figure supplement 1*) as described previously (*Roschger et al., 1998*). Five parameters were derived to characterize the BMDD (*Roschger et al., 2008*). For information about $Ca_{Young}$, which is the mean calcium concentration of the bone area between the double fluorescence labels, the identical bone surface measured with qBEI was additionally imaged in a Confocal Laser Scanning Microscope (Leica TCS SP5, Leica Microsystems CMS GmbH, Wetzlar, Germany) using a laser light of 405 nm for fluorescence excitation and a 20x object lens (pixel resolution of 0.76 μm). By matching the CLSM with the qBEI images, the sites of the fluorescence labels were overlaid exactly onto the qBEI images (*Figure 1—figure supplement 1E*). $Ca_{Young}$ was obtained from a total of 39 areas from a subgroup of nine samples and was subsequently used to calculate $Ca_{Low}$, which reflects the percentage of newly formed bone area (*Figure 1—figure supplement 1H*).

## Flow cytometry

Ubiquitin-Cre$^{ERt2}$; tdTomato$^{LSL}$ mice were treated with tamoxifen (1 mg IP Q48H, three injections total). Two weeks after the first tamoxifen injection, mice were sacrificed and bone marrow cells were isolated by flushing with ice cold PBS using a 25G needle. $10^6$ bone marrow cells were protected from light and stained on ice for 30 min with the following primary antibodies in a 10 μL staining volume. APC anti-mouse/human CD 11b antibody (Biolegend, San Diego, CA), FITC anti-mouse CD3 antibody (Biolegend, San Diego, CA), and PE/Cy7 anti-mouse Ly-6C antibody (Biolegend, San Diego, CA). After staining, cells were washed twice with FACS buffer (PBS plus 2% heat inactivated fetal bovine serum) and analyzed on a SORP 8 Laser BD LSR flow cytometer (Becton, Dickinson and Company, Franklin Lakes, NJ).

## Bone marrow macrophages and osteoclast differentiation

Bone marrow cells were harvested from 4 week-old C57BL/6J mice. Femur and tibia were removed and the marrow cavity was flushed out with 10 mL 1x phosphate-buffered saline per mouse (GE Healthcare Life Sciences, MA) with a 25 G needle (Becton, Dickinson and Company, (Franklin Lakes, NJ)) which was then centrifuged at 1,000 rpm for 10 min. Supernatant was removed, and cell pellet was resuspended in 1 mL red blood cell lysis buffer (Sigma-Aldrich, St. Louis, MO) and incubated for 2 min followed by centrifugation at 1000 rpm for 5 min to collect cells. Bone marrow cells were plated on 100 mm non-treated tissue culture dishes (Corning Inc, Corning, NY) and maintained in alpha minimum essential medium eagle (MEM) (Sigma-Aldrich, St. Louis, MO) containing 15% fetal bovine serum FBS (Gemini Bio, West Sacramento, CA), 1x penicillin-streptomycin (Gibco, Waltham, MA), 1x GlutaMAX supplement (Gibco, Waltham, MA) and in the presence of 30 ng/mL recombinant mouse M-CSF (R and D systems, Minneapolis, MN). These cells were maintained at 37°C in a humidified 5% $CO_2$ incubator for 3 days, then collected by trypsinization. Primary osteoclast precursors were then seeded in 96 well plates at 2,000 cells/well (20,000 cells/mL), and differentiation was induced with 50 ng/mL of recombinant mouse RANKL (R and D systems, Minneapolis, MN) and 30

ng/mL recombinant mouse M-CSF (R and D systems, Minneapolis, MN). Fresh medium was replenished every 3 days.

For experiments with 4-hydroxytamoxifen treatment, primary osteoclast precursors were seeded after initial culture on non-treated plates into six-well plates in the presence of 30 ng/mL recombinant mouse M-CSF (R and D systems, Minneapolis, MN) for 3 days. Cells were then treated with 300 nM of 4-hydroxytamoxifen (Sigma-Aldrich, St. Louis, MO) or vehicle (methanol, Sigma-Aldrich, St. Louis, MO) for 3 days. Fresh medium with 30 ng/mL recombinant mouse M-CSF (R and D systems, MN) was then added for another 3 days prior to collecting cells for subsequent osteoclast differentiation.

For osteoclast differentiation assays, primary osteoclast precursors were seeded in 96-well plates, and differentiation was induced with 50 ng/mL of recombinant mouse RANKL (R and D systems, Minneapolis, MN) and 30 ng/mL recombinant mouse M-CSF (R and D systems, Minneapolis, MN). Fresh medium was replenished every 3 days. sTRAP assay was measured with acetate buffer from acid phosphatase leukocyte kit (Sigma-Aldrich, St. Louis, MO), 1M sodium L-tartrate dibasic dihydrate (Sigma-Aldrich, St. Louis, MO) and pNPP substrate (Sigma-Aldrich, St. Louis, MO). Fifty µL of cell culture supernatant was transferred to a new 96-well plate and 150 µL of substrate mix was added. Plate was then incubated 37°C for an hour. Reaction was terminated by adding 3N NaOH (Sigma-Aldrich, St. Louis, MO), which results in an intense yellow color. Absorbance was measured at 405 nm.

TRAP staining was performed using acid phosphatase leukocyte kit (Sigma-Aldrich, St. Louis, MO) to visualize osteoclasts. TRAP buffer was prepared freshly on the day of experiment by mixing acetate buffer containing sodium acetate and acetic acid with sodium tartrate and naphthol AS-BI phosphate disodium salt (Sigma-Aldrich, St. Louis, MO). TRAP staining solution was prepared by mixing Fast Garnet GBC Base with sodium nitrite. For TRAP staining, culture medium was removed from cells, and cells were fixed with fixative solution for 5 min at room temperature. Fixative solution was prepared fresh on the day of experiment by mixing citrate solution, acetone with 37% formaldehyde. Cells were then washed twice with deionized water and stained with TRAP staining solution for 60 min at 37°C.

For pit resorption assays, primary osteoclast precursors were seeded in 96-well resorption plates at 2000 cells/well (Corning, Inc, Corning, NY), and differentiation was induced with 50 ng/mL of recombinant mouse RANKL and 30 ng/mL recombinant mouse M-CSF. Fresh medium with M-CSF and RANKL was replenished every 3 days. Culture was aspirated and cells were washed three times with 1x PBS. A total of 100 µL of 10% bleach was added to each well and incubated for 30 min at room temperature. Bleach solution was removed and cells were washed twice with deionized water. Plates were air-dried at 4°C overnight and osteoclast resorption pits were visualized the next day. Pit resorption was quantified using ImageJ.

## gDNA deletion analysis

Genomic DNA was isolated from cultured bone marrow macrophages or cortical bone using DNeasy Blood and Tissue Kit (Qiagen, Hilden, Germany) according to the instructions of the manufacturer. For cultured bone marrow macrophages, cell were treated with vehicle (methanol) or 4-hydroxytamoxifen (0.3 µM) for 72 hr prior to gDNA isolation. For cortical bone, mice were treated with tamoxifen (three 1 mg IP injection every 48 hr) and then sacrificed 7 days after the first tamoxifen injection. Epiphyses were removed and bone marrow cells were flushed using ice cold PBS. Remaining cortical bone fragments were flash frozen in liquid nitrogen and then pulverized using a mortar and pestle. Bone tissue was then subjected to gDNA isolation and quantified using a spectrophotometer (Nanodrop 2000, Thermo Fisher Scientific, Waltham, MA). 15 ng gDNA was used for each qPCR reaction. For each gene, one primer pair was used that was internal to the targeted loxP sites and a second external primer pair was used to normalize to input gDNA amount. See PCR primer Appendix for primer sequences used. Relative abundance of the targeted gene was calculated by the 2(-Delta Delta C(T)) method using the external primer pair as control (*Livak and Schmittgen, 2001*).

## TF-seq

Ocy454 cells were infected with lentiviral particles expressing TF-seq library (*O'Connell et al., 2016*) and eGFP. Infected cells were isolated by sorting for GFP expression. The cell line was confirmed to

be mycoplasma-free by PCR. Thereafter, cells were plated at a density of 50,000/ml in 96 well plates (5000 cells/well) and allowed to expand at 33°C for 48 hr. Cells were then moved to 37°C for 24 hr, then treated with vehicle (DMSO) or YKL-05–099 for time points ranging from zero minutes to 4 hr with n = 3 wells per condition. Cells were washed with ice cold PBS and then lysed with RLT buffer (Qiagen, supplemented with ß-mercaptoethanol) and stored at −80°C. RNA was purified using Agentcourt RNA Clean XP to precipitate the nucleic acids in 1.25 M NaCl and 10% PEG-8000. Maxima reverse transcriptase (Thermo Fisher Scientific) was run according to the manufacturer's instructions using a multiplexed primed reverse transcriptase reaction. The biotinylated 96-well sequence tagged TF-seq-specific reverse transcriptase primers, and the biotinylated 96-well degenerate sequence-tagged polydT reverse transcriptase primers were used at 750 and 250 nM final concentrations, respectively, with 50 units of Maxima. After sequencing-tagging all cDNA during reverse transcriptase, each 96-well plate was pooled and the unincorporated primers were washed away from the cDNA by precipitating with 10% PEG-8000 and 1.25 M NaCl. Amplification of the TF-seq gene reporter amplicon was performed on 50% of the cDNA using primers with full Illumina-compatible sequencing adapters in a 600 µl PCR reaction for 28 cycles. The 422 bp amplicon was then gel extracted for sequencing. TF-seq is a 50 bp single-end read, well-tag, RNA UMI, followed by a 17 bp constant sequence, then the reporter tag UMI, and ultimately the reporter tag. We counted the number of unique RNA molecules for every well and reporter element, requiring a perfect match for the respective tags. UMI tag counts for each reporter element were obtained. Reporter element activity was expressed as fold change versus baseline (time 0) for each drug treatment.

## Generation of models of kinases in various conformational states

Models of CSF1R (FMS), SIK1, and SIK2 (QIK) in four conformations, the CIDI, CIDO, CODI, and CODO states, were generated using DFGmodel (*Ung and Schlessinger, 2015*). DFGmodel uses a multi-template homology modeling approach to construct composite homology models in various kinase conformations. The models are based on an augmented version of a structurally validated sequence alignment of human kinome (*Modi and Dunbrack, 2019*). For the CIDO, CODI, and CODO states, DFGmodel uses multi-template approach, where manually curated sets of determined structures, covering a unique range of conformations for each state are used as template structures. For the CIDI, DFGmodel uses a single-template approach to construct the models; DFGmodel searches a library of kinase structures annotated by Kinformation (*Ung et al., 2018*; *Rahman et al., 2019*; *Camacho et al., 2009*; http://www.kinametrix.com) to identify a best-matched kinase in CIDI state as template. DFGmodel then uses MODELLER (*Sali and Blundell, 1993*) v9.21 to generate homology models for each kinase. For CIDI, CIDO, and CODO states, 50 models were generated; POVME (*Durrant et al., 2014*) v2.1 was used to estimate the binding site volume of each model, where 10 models with the largest volume were selected. For CODI, four sets of templates are required to cover the observed conformation, thus 40 models were generated for each of these sets; eight models with the largest volume in each of the sets were selected. Lastly, CSF1R crystal structures are also included for examination.

## Molecular docking

Molecular docking was performed using Glide (*Friesner et al., 2004*) (Schrödinger 2019–3). Default settings and Standard Precision (SP) model, with the addition of aromatic hydrogen and halogen bonds, were used for Glide protein grid generation and docking of inhibitor into the active site of protein kinase models in CIDI, CIDO, CODI, and CODO conformations. OPLS3e force field (*Harder et al., 2016*) was used to parameterize both protein and ligands. Scaling of van der Waals (vdW) radius for receptor atoms was set to 0.75. The docking results were averaged for 10 models, as described by *Ung and Schlessinger, 2015*. The small molecule YKL-05–099 was prepared with Schrödinger's LigPrep program. Docking pose of this molecule in each of the models was examined; poses that do not possess the critical hydrogen-bonds with the hinge residues or have docking score higher than −7.0 were rejected.

## CSF1R kinase assay

Assays were performed in base reaction buffer (20 mM Hepes (pH 7.5), 10 mM MgCl2, 1 mM EGTA, 0.02% Brij35, 0.02 mg/ml BSA, 0.1 mM Na3VO4, 2 mM DTT, 1% DMSO). YKL-05–099 was dissolved

in 100% DMSO in a 10 mM stock. Serial dilution was conducted by Integra Viaflo Assist in DMSO. Recombinant CSF1R (Invitrogen, Carlsbad, CA) was used at a concentration of 2.5 nM. The substrate used was pEY (Sigma-Aldrich, St. Louis, MO) at a concentration of 0.2 mg/ml. Kinase assays were supplemented with 2 mM $Mn^{2+}$, and 1 μM ATP was added. Assays were performed for 20 min at room temperature, after which time $^{33}$P-ATP (10 μCi/μl) was added followed by incubation for another 120 min at room temperature. Thereafter, radioactivity incorporated into the pEY peptide substrate was detected by filter-binding method. Kinase activity data were expressed as the percent remaining kinase activity in test samples compared to DMSO reactions. $IC_{50}$ values and curve fits were obtained using Prism 8.0 GraphPad Software (GraphPad Software, San Diego, CA).

## Immunoblotting

Immunoblotting was performed in lysates derived from primary bone marrow macrophages. Cells were scraped into ice cold PBS and cell pellets were lysed in TNT (200 mM NaCl, 20 mM Tris HCl pH 8, 0.5% Triton X-100 supplemented with protease and phosphatase inhibitor cocktail (Thermo Fisher Scientific, Waltham, MA)). Cells were vortexed twice in lysis buffer for 30 s followed by centrifugation at top speed at 4°C for 5 min. Protein concentration in cell lysates was quantified using Bradford assay (Thermo Fisher Scientific, Waltham, MA). Equal amounts of protein were then separated by SDS-PAGE under reducing conditions. Proteins were then transferred to nitrocellulose membranes. Membranes were blocked in TBST plus 5% milk for 30 min at room temperature. Thereafter, membranes were incubated overnight at 4°C in TBST with 5% BSA plus primary antibodies (source and dilutions below). Membranes were then washed three times with TBST for 5 min, followed by secondary antibody incubation (1:2000 goat anti-rabbit) for 30 min at room temperature. Next, membranes were washed again for three times with TBST for five minutes followed by detection with Pierce ECL Plus Western Blotting Substrate (Thermo Fisher Scientific, Waltham, MA) and imaging by Azure c600. The primary antibodies were: SIK2 (1:1000), CSF1R phospho-Y723 (1:1000), CSF1R (1:1000), phospho-ERK1/2 (1:1000), ERK1/2 (1:2000), Tubulin (1:1000) were obtained from Cell Signaling Technology (Danvers, MA). Anti-SIK3 rabbit polyclonal antibodies (1:1000) were obtained from Abcam (Cambridge, United Kingdom).

## Statistical analysis

When two groups were compared, statistical analyses were performed using unpaired two-tailed Student's t-test. Population samples were shown to be normally distributed. For OVX studies (*Figures 1* and *2*), two-way ANOVA was performed followed by Dunnett's multiple comparisons test (GraphPad Prism 9.1.1). p Values less than 0.05 were considered to be significant. The numbers of mice studied in all experiments are described in figure legends, and in all figures data points represent individual mice. All data points indicate individual biologic replicates (independent experimental samples) and not technical replicates (the same sample re-analyzed using the same method). A Source Data File is presented for all figures.

## Acknowledgements

We thank Drs. Michael Mannstadt, Lauren Surface, Francesca Gori, Tatsuya Kobayashi, Mark Poznansky, Christiana Iyasere, and members of the Wein laboratory for helpful discussions. MNW acknowledges funding support from the American Society of Bone and Mineral Research, the Harrington Discovery Institute, the MGH Department of Medicine Innovation Program, the Gillian Reny Stepping Strong Center for Trauma Innovatio, and the National Institute of Health (DK116716, AR066261, and AR067285). HMK acknowledges funding support from the National Institute of Health (AR066261 and DK011794). MF acknowledges funding support from Centre National de la Recherche Scientifique (CNRS), the Socieťeˊ Francophone du Diabète (SFD) and the Fondation pour la Recherche Meˊdicale (FRM). RB acknowledges funding support from the National Institutes of Health (DK092590 and AR059847).

# Additional information

## Competing interests

Nathanael S Gray, Thomas B Sundberg, Ramnik J Xavier: co-inventor on a pending patent (US Patent Application 16/333,546) regarding the use of SIK inhibitors for osteoporosis. Peter Ung: is currently affiliated with Genentech. The author has no financial interests to declare. Henry M Kronenberg: co-inventor on a pending patent (US Patent Application 16/333,546) regarding the use of SIK inhibitors for osteoporosis. Receive research support from Galapagos NV. Marc N Wein: co-inventor on a pending patent (US Patent Application 16/333,546) regarding the use of SIK inhibitors for osteoporosis. Receives research support from Radius Health. Receive research support from Galapagos NV. The other authors declare that no competing interests exist.

## Funding

| Funder | Grant reference number | Author |
|---|---|---|
| National Institute of Arthritis and Musculoskeletal and Skin Diseases | AR066261 | Marc N Wein |
| National Institute of Diabetes and Digestive and Kidney Diseases | DK116716 | Marc N Wein |
| National Institute of Arthritis and Musculoskeletal and Skin Diseases | AR067285 | Marc N Wein |
| National Institute of Diabetes and Digestive and Kidney Diseases | DK011794 | Henry M Kronenberg |
| National Institute of Diabetes and Digestive and Kidney Diseases | DK092590 | Rebecca Berdeaux |
| National Institute of Arthritis and Musculoskeletal and Skin Diseases | AR059847 | Rebecca Berdeaux |

The funders had no role in study design, data collection and interpretation, or the decision to submit the work for publication.

## Author contributions

Cheng-Chia Tang, Conceptualization, Data curation, Formal analysis, Investigation, Methodology, Writing - original draft, Writing - review and editing; Christian D Castro Andrade, Maureen J O'Meara, Janaina da Silva Martins, Investigation, Methodology, Writing - review and editing; Sung-Hee Yoon, Resources, Investigation, Methodology, Writing - review and editing; Tadatoshi Sato, Thomas B Sundberg, Peter Ung, Formal analysis, Investigation, Methodology, Writing - review and editing; Daniel J Brooks, Data curation, Methodology, Writing - review and editing; Mary L Bouxsein, Formal analysis, Supervision, Investigation, Methodology, Writing - review and editing; Jinhua Wang, Nathanael S Gray, Conceptualization, Resources, Methodology, Writing - review and editing; Barbara Misof, Conceptualization, Formal analysis, Methodology, Writing - original draft, Writing - review and editing; Paul Roschger, Resources, Formal analysis, Writing - review and editing; Stephane Boulin, Data curation, Formal analysis, Writing - review and editing; Klaus Klaushofer, Resources, Formal analysis, Supervision, Writing - review and editing; Annegreet Velduis-Vlug, Daniel O'Connell, Formal analysis, Investigation, Writing - review and editing; Yosta Vegting, Formal analysis, Writing - review and editing; Clifford J Rosen, Formal analysis, Supervision, Writing - review and editing; Ramnik J Xavier, Supervision, Methodology, Writing - review and editing; Avner Schlessinger, Formal analysis, Supervision, Investigation, Methodology; Henry M Kronenberg, Conceptualization, Writing - review and editing; Rebecca Berdeaux, Marc Foretz, Conceptualization, Resources, Writing - review and editing; Marc N Wein, Conceptualization, Data curation, Formal analysis,

Supervision, Funding acquisition, Investigation, Methodology, Writing - original draft, Project administration, Writing - review and editing

## Author ORCIDs
Daniel J Brooks https://orcid.org/0000-0001-7408-9851
Nathanael S Gray https://orcid.org/0000-0001-5354-7403
Avner Schlessinger https://orcid.org/0000-0003-4007-7814
Marc N Wein https://orcid.org/0000-0002-6015-8147

## Ethics
Animal experimentation: All procedures involving animals were performed in accordance with guidelines issued by the Institutional Animal Care and Use Committees (IACUC) in the Center for Comparative Medicine at the Massachusetts General Hospital and Harvard Medical School under approved Animal Use Protocols (2019N000201).

## Additional files

### Supplementary files
- Source data 1.
- Supplementary file 1. µCT results from ovariectomy study.
- Supplementary file 2. Mechanical testing results from ovariectomy study.
- Supplementary file 3. Histomorphometry measurements from ovariectomy study.
- Supplementary file 4. Quantitative backscatter electron imaging results from ovariectomy study.
- Supplementary file 5. µCT results from control and *Sik2/3* DKO mice.
- Supplementary file 6. Histomorphometry measurements from control and *Sik2/3* DKO mice.
- Supplementary file 7. Sequence of PCR primers.
- Transparent reporting form

### Data availability
Source data files have been provided for all figures.

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
