## [Decision Letter]

**Acceptance summary:**

The paper shows very clearly that dual targeting of SIK2/3 and CSF1R by a novel small molecule induces bone formation without increasing bone resorption. This novel strategy may overcome limitations of most current anabolic osteoporosis therapies, such as PTH. The reviewers agree that their comments and concerns were addressed.

**Decision letter after peer review:**

Thank you for submitting your article "Dual targeting of salt inducible kinases and CSF1R uncouples bone formation and bone resorption" for consideration by *eLife*. Your article has been reviewed by 3 peer reviewers, one of whom is a member of our Board of Reviewing Editors, and the evaluation has been overseen by Mone Zaidi as the Senior Editor. The reviewers have opted to remain anonymous.

All three reviewers and the Editors agree this is a very interesting, novel and informative study. However, a few issues need to be addressed to strengthen its conclusions and its biological and translational relevance.

Essential Revisions:

1. Please address point-by-point reviewers' concerns and comments.

2. In particular, please explain why the pharmacological studies were performed at 24 weeks of age whereas the genetic models were analyzed at 9 weeks. Alternatively, perform additional experiments as suggested by Reviewer #1.

*Reviewer #1 (Recommendations for the authors):*

The primary objective of this manuscript was to examine if multi-kinase inhibitor YKL-05-099 can inhibit salt inducible kinases (SIKs) with the goal to examine a new class of bone anabolic agents for the treatment of osteoporosis. The hypothesis tested is that deletion/inhibition of SIK2 and SIK3, broadly-expressed AMPK family serine/threonine kinases, would increase trabecular bone mass and other outcome measures associated with diminished bone quality and density seen in osteoporosis. To this end, the authors provide a thorough investigation of their hypotheses using robust measurements and validate their findings utilizing multiple techniques. They tested YKL-05-099 in female mice rendered hypogonadal by surgical oophorectomy. Their outcome measures of interest were trabecular bone mass, bone formation, and bone resorption and found that YKL-05-099 was successful in increasing anabolism and, surprisingly, decreasing resorption, leading them to investigate why this inhibitor differed from the effects of deletion of SIK2 and SIK3. They found that YKL-05-099 also inhibited the CSF1 (M-CSF) receptor, thus, inhibiting osteoclast activity. This is an interesting manuscript but there are some flaws in the conduct of the experiments and in the analyses which lessen its impact.

1. The authors initially recognize that PTH-derived osteoanabolic agents are limited by concomitant stimulation of bone resorption in the introduction but do not address this limitation in detail with sufficient background/context in the discussion/conclusions.

2. It might also be interesting to address recent clinical trials on denosumab (RankL inhibitor) and how inhibition of SIKs might affect downstream RankL production.

3. They need to specify that the mice are C57Bl/6J for the ovariectomy studies, even though they mention that the wild type mice were purchased from Jax. Also, they do not give the background strains of the genetic models. They must do that, and if they are not C57Bl/6J, it is difficult to make a comparison between the studies. They say they use both male and female mice, but only provide data for females. This is reasonable, but where do they use male mice?

4. They have a problem with their power analyses, or lack of them. They used a published model to determine their sample size for the ovariectomy studies, but did not achieve the endpoint of the published study (femoral trabecular BV/TV decreasing from 7.8% to 4.2%). In fact, they had no difference in the femoral trabecular BV/TV and barely significant difference in tBMD with ovariectomy. Nevertheless, they did observe a highly significant decrease in the vertebral BV/TV. But it illustrates that they needed more mice in the ovariectomy studies, more like 12/group. They even state that "since more stringent ANOVA analysis will be required… a larger size will be necessary" (I will come back to the statistical analyses below, which are also not correct). In addition, in Figure 1., only the cortical thickness seems to show 8 animals/group. In fact, in the L5 BV/TV the PTH-treated groups are only 4 and 5 animals, as far as I can tell. Why are there less than 8 animals/group in some of the panels, and how did they decide to exclude, or not analyze, some of the animals? With respect to the genetic gene deletion studies, why did they not perform power analyses prior to undertaking the experiments?

5. They have to perform 2-way ANOVA for all of the ovariectomy work. They cannot use 1-way ANOVA. They also must show all their data as means + SD, not SEM. Where they used Student's t-tests with 2 groups, were the population samples proven to be normally distributed? Would the significance remain the same if non-parametric tests were used?

6. With respect to Figure 1., they have to modify the text in the Results. They cannot state "that YKL-05-099 tended to greater gains in trabecular bone mass". It's about the same as for PTH. They do not show that there is any significant difference between PTH and YKL-05-099 in the OVX mice. They also need to add to this sentence "in these mice but had no effect on intact mice". This is an interesting observation, and suggests that YKL-05-099 is only effective when there is an increase in osteoclast activity.

7. Notwithstanding, there are some issues with Figure 2., that do not fit with Figure 1. Why is the BFR so high in the Sham-YKL-05-099-treated mice, when there is no increase in BV/TV in these mice? Also, they must add osteoclast numbers to the figure. This is imperative.

8. They have to be careful with the use of the term "bone mass" that they are really meaning "mass" not "area" or "volume". For instance, when referring to the Nishimori paper on p. 6, second paragraph, "area" may be more appropriate for the massive changes that occur in development with deletion of the Siks. Similarly, with the uCT and histomorphometry in Figures 3 and 4. In fact, the uCT images in Figure 3B appear to show a great increase in the growth plate or resorption (it is hard to tell). It is most important that they indicate where in the trabecular bone the uCT analysis was done in each of the groups of mice.

9. The age of the DKO mice (6-9 weeks) used in Figures 3 and 4 is not appropriate to compare with the YKL-05-099-treated OVX mice (24 weeks old). The DKO mice are young, rapidly growing mice, as they note on p.6, 2nd paragraph. But then, they refer to them as adult-onset DKO in the next paragraph, and in the title of the legends. This is not correct; the mice will have just reached puberty at 6 weeks when the inducible DKO occurs. They must repeat this study with mice the same age as the OVX mice. They may obtain a milder effect of SIK deletion on the bone phenotype.

10. Why did they examine the metabolic parameters at 8 weeks of age, not 9, which is when they killed the other mice? Also, were all the mice fasted for the same 3 h? What time is ZT3? Perhaps the increase in blood glucose and BUN caused by YKL-05-099 were because of the age differences?

11. In Figure 5, it seems that the 4-OHT has effects on the abundance of SIK2 and SIK3, and there is less protein in vehicle-treated DKO cells, suggesting the Cre is leaky. They also need to show the abundance of SIK1.

12. Figure 7. shows that YKL-05-099 may inhibit a number of tyrosine kinases. They need to address the possibility that some of the effects they observe are due to other kinases than SIKs and the CSF1R. The effect on the growth plate may be similar to that of gefitinib and the off-target effects could be due to inhibition of other kinases.

13. Why did they use such a high dose (15 uM) of YKL-05-099 in Figure 8E, F and G? This might be toxic.

14. How many times were the in vitro experiments repeated. This needs to be mentioned in the methods or legends.

*Reviewer #2 (Recommendations for the authors):*

The authors should present osteoclast surface or number in figure 2D because ES/BS can be affected by changes in bone formation. Specifically, an increase in bone formation can reduce eroded surface even if osteoclast number does not change.

*Reviewer #3 (Recommendations for the authors):*

In this study, Tang and colleague report that the multikinase inhibitor YKL-05-099 increases bone formation and decreases bone resorption in hypogonadal female mice with mechanisms that are likely to involve inhibition of SIKs and CSFR1, respectively. The authors also report that postnatal mice with inducible, global deletion of SIK2 and SIK3 show an increase of bone mass that is associated to both an augmentation of bone formation and bone resorption.

The paper provides novel and interesting information with potentially highly relevant translational implications. The quality of the data is outstanding and most of the authors' conclusions are supported by the data as shown.

1. Measurements of osteoclast number upon pharmacological treatments should be provided.

2. What is the experimental evidence that the increase of bone formation observed in mice treated with YKL-05-099 is mediated by SIKs inhibition? It would be helpful if the authors could discuss this point in the Discussion section.

---

## [Author Response]

Essential Revisions:1. Please address point-by-point reviewers' concerns and comments.2. In particular, please explain why the pharmacological studies were performed at 24 weeks of age whereas the genetic models were analyzed at 9 weeks. Alternatively, perform additional experiments as suggested by Reviewer #1.Reviewer #1 (Recommendations for the authors):The primary objective of this manuscript was to examine if multi-kinase inhibitor YKL-05-099 can inhibit salt inducible kinases (SIKs) with the goal to examine a new class of bone anabolic agents for the treatment of osteoporosis. The hypothesis tested is that deletion/inhibition of SIK2 and SIK3, broadly-expressed AMPK family serine/threonine kinases, would increase trabecular bone mass and other outcome measures associated with diminished bone quality and density seen in osteoporosis. To this end, the authors provide a thorough investigation of their hypotheses using robust measurements and validate their findings utilizing multiple techniques. They tested YKL-05-099 in female mice rendered hypogonadal by surgical oophorectomy. Their outcome measures of interest were trabecular bone mass, bone formation, and bone resorption and found that YKL-05-099 was successful in increasing anabolism and, surprisingly, decreasing resorption, leading them to investigate why this inhibitor differed from the effects of deletion of SIK2 and SIK3. They found that YKL-05-099 also inhibited the CSF1 (M-CSF) receptor, thus, inhibiting osteoclast activity. This is an interesting manuscript but there are some flaws in the conduct of the experiments and in the analyses which lessen its impact.1. The authors initially recognize that PTH-derived osteoanabolic agents are limited by concomitant stimulation of bone resorption in the introduction but do not address this limitation in detail with sufficient background/context in the discussion/conclusions.

This is an excellent point. We have re-emphasized this concept in the first paragraph of the revised discussion along with new references to important review articles on this topic:

“Currently, analogs of parathyroid hormone (teriparatide and abaloparatide) represent the most commonly-used class of bone anabolic osteoporosis treatment agents. While effective in increasing spine bone density, efficacy of such agents may be limited by concomitant stimulation of bone resorption (5). For this reason, novel strategies that stimulate bone formation without simultaneous induction of bone resorption are highly desired (47).”

With these changes, appropriate context is now provided for readers to appreciate the potential clinical significance of this work.

2. It might also be interesting to address recent clinical trials on denosumab (RankL inhibitor) and how inhibition of SIKs might affect downstream RankL production.

Again, we appreciate this opportunity to provide additional clinical context for this study and to comment on the relationship between SIKs and RANKL expression. Since anti-resorptive agents (bisphosphonates and denosumab) represent first line osteoporosis therapy, we have revised the discussion to first state:

“Anti-resorptive therapies (bisphosphonates and the anti-RANKL antibody denosumab) have long represented first line treatment for patients with osteoporosis. These agents boost bone mass and reduce fractures (47), yet are linked to rare but serious side effects (osteonecrosis of the jaw and atypical femur fractures) which are related to excessive suppression of both bone resorption and bone formation (48).”

Next, we included a sentence in the second paragraph of the discussion to remind the reader about the relationship between SIKs and control of RANKL expression:

“High bone resorption seen in response to Sik2/3 deletion is likely driven by increased RANKL expression by osteoblasts and osteocytes (9).”

3. They need to specify that the mice are C57Bl/6J for the ovariectomy studies, even though they mention that the wild type mice were purchased from Jax. Also, they do not give the background strains of the genetic models. They must do that, and if they are not C57Bl/6J, it is difficult to make a comparison between the studies. They say they use both male and female mice, but only provide data for females. This is reasonable, but where do they use male mice?

We thank the reviewer for bringing up these important points. The Methods section notes that C57Bl/6J mice from JAX (catalog number 000664) were used for OVX studies. The Results text has been revised to include this information as well. The methods section has been revised to indicate that are genetically-modified mice have been back-crossed to C57Bl/6J for at least 5 generations. The figure legends have been revised to indicate where male versus female mice were used for our gene deletion studies. We observe comparable (and dramatic) skeletal effects in response to *Sik2/3* deletion in male and female mice. Fasting serum analyses were performed with male mice while histology and microCT was performed on female mice.

4. They have a problem with their power analyses, or lack of them. They used a published model to determine their sample size for the ovariectomy studies, but did not achieve the endpoint of the published study (femoral trabecular BV/TV decreasing from 7.8% to 4.2%). In fact, they had no difference in the femoral trabecular BV/TV and barely significant difference in tBMD with ovariectomy. Nevertheless, they did observe a highly significant decrease in the vertebral BV/TV. But it illustrates that they needed more mice in the ovariectomy studies, more like 12/group. They even state that "since more stringent ANOVA analysis will be required… a larger size will be necessary" (I will come back to the statistical analyses below, which are also not correct). In addition, in Figure 1., only the cortical thickness seems to show 8 animals/group. In fact, in the L5 BV/TV the PTH-treated groups are only 4 and 5 animals, as far as I can tell. Why are there less than 8 animals/group in some of the panels, and how did they decide to exclude, or not analyze, some of the animals? With respect to the genetic gene deletion studies, why did they not perform power analyses prior to undertaking the experiments?

We thank the reviewer for these thoughtful and important questions. Prior to performing the OVX studies in Figures 1 and 2, we performed power calculations which suggested a sample size of n=7/group. As detailed in the Methods section and above, we chose a sample size of n=8/group. While our study did not reveal obvious OVX-induced bone loss in the trabecular compartment of the distal femur metaphysis, we did observe clear OVX-induced bone loss in trabecular bone at L5. This experience highlights the common challenge to demonstrate trabecular bone loss in mouse OVX studies, especially in strains like C57Bl/6J where baseline levels of femur metaphyseal trabecular bone are low. For this reason, future studies will be required using other strains or rat models which may be more robust to detect OVX-induced bone loss. The Discussion (3^rd^ to last paragraph) has been revised to revised to state:

“A final limitation of this OVX study is that our model using C57Bl/6J mice showed obvious OVX-induced loss of trabecular bone volume fraction (BV/TV) in L5 vertebrae but not in the distal femur metaphyseal region (Figure 1C, E). Demonstrating OVX-induced loss of trabecular bone in the appendicular skeleton can be challenging, especially in strains (like C57Bl/6J) which show low amounts of trabecular bone at this skeletal site (64). Future studies will be needed in other mouse strains and/or robust rat OVX models (65) to further demonstrate the therapeutic efficacy of SIK inhibitors in bone loss associated with hypogonadism.”

We were unable to process some samples due to technical problems associated with bone dissection at the time of necropsy (growth plate trans-section leading to inability to measure distal femur parameters and vertebral body damage and/or isolation of the incorrect vertebral body). The Methods section has been revised to indicate this. No measured samples were excluded from analysis.

For our gene deletion studies, no power calculations were performed prior to starting experiments, as is frequently the case in skeletal phenotypic description of genetically-modified mice. The phenotype observed with postnatal ubiquitous *Sik2/3* deletion is quite dramatic and consistent across mice analyzed.

5. They have to perform 2-way ANOVA for all of the ovariectomy work. They cannot use 1-way ANOVA. They also must show all their data as means + SD, not SEM. Where they used Student's t-tests with 2 groups, were the population samples proven to be normally distributed? Would the significance remain the same if non-parametric tests were used?

We thank the reviewer for raising these important points. We obtained a biostatistics consultation from the Harvard Catalyst program to review our statistical methods in detail. For comparison between 2 groups using Student’s t-tests, population samples were normally distributed. The Methods section has been revised to include this information. For the ovariectomy study, we initially used the default analysis method in our statistical software program (GraphPad Prism 9.1.1) for overall comparisons and then post-hoc between group analysis. The default method actually was a 2-way ANOVA method followed by Dunnett’s multiple comparisons tests. The Methods section and figure legends have revised to include this important information. We apologize for any confusion. Finally, we have revised all figures to show data as means ± SD.

6. With respect to Figure 1., they have to modify the text in the Results. They cannot state "that YKL-05-099 tended to greater gains in trabecular bone mass". It's about the same as for PTH. They do not show that there is any significant difference between PTH and YKL-05-099 in the OVX mice. They also need to add to this sentence "in these mice but had no effect on intact mice". This is an interesting observation, and suggests that YKL-05-099 is only effective when there is an increase in osteoclast activity.

We thank the reviewer for raising these excellent points. While the numeric means of the trabecular bone changes in response to YKL-05-099 were higher than those seen following PTH treatment, we agree that these differences are minor, and the text has been revised as suggested since the comparison between OVX+YKL and OVX+PTH was not statistically significant in any instances. Based on the observation that YKL-05-099 may be more effective in OVX mice (with increased osteoclast activity), we have revised the Discussion (second paragraph) to include the sentence:

“Interestingly, YKL-05-099 appears to show greater efficacy with respect to increasing trabecular bone mass in OVX mice, suggesting that this compound relies on states of high osteoclast activity for its full bone-building potential.”

7. Notwithstanding, there are some issues with Figure 2., that do not fit with Figure 1. Why is the BFR so high in the Sham-YKL-05-099-treated mice, when there is no increase in BV/TV in these mice? Also, they must add osteoclast numbers to the figure. This is imperative.

These are important questions, and we appreciate the opportunity to respond. In the initial manuscript, osteoclast numbers from the OVX histomorphometry study were shown in Supplemental Table 3. Based on the suggestion from all three reviewers, we have revised Figure 2 to include graphs of these results along with eroded surface data. Both pieces of histomorphometry data show a similar message: osteoclast parameters tend to increase with PTH (as expected) and decrease with YKL-05-099 treatment. We agree that the potential disconnect between BFR/BS (histomorphometry, Figure 2H) and BV/TV (microCT, Figure 1C) data for the Sham+YKL group is interesting. 4 weeks of drug treatment may not be sufficient time to see changes in bone matrix deposition (measured by histomorphometry) ‘translate’ into increased trabecular bone by microCT. In addition, we note that there is a numerical increase in BV/TV in the Sham+YKL group compared to Sham+VEH (p=0.077). Therefore, the histomorphometry and microCT data here are internally consistent.

8. They have to be careful with the use of the term "bone mass" that they are really meaning "mass" not "area" or "volume". For instance, when referring to the Nishimori paper on p. 6, second paragraph, "area" may be more appropriate for the massive changes that occur in development with deletion of the Siks. Similarly, with the uCT and histomorphometry in Figures 3 and 4. In fact, the uCT images in Figure 3B appear to show a great increase in the growth plate or resorption (it is hard to tell). It is most important that they indicate where in the trabecular bone the uCT analysis was done in each of the groups of mice.

This is a very important point. Indeed, *Sik2/3* deletion (with Dmp1-Cre or ubiquitin-Cre^ERt2^) causes increased trabecular bone area (presumably due cortical remodeling) and trabecular bone volume fraction (BV/TV, often referred to as trabecular bone mass) due to increased bone formation. We have revised the sentence on Page 6 accordingly. As indicated on page 6, we note growth plate expansion in response to Sik2/3 deletion with ubiquitin-CreERt2 (Figure 3B, 3E, 4A, and S5) along with dramatic changes in trabecular bone. The Methods section details how the 0.2 mm distal metaphyseal region was selected for trabecular bone micro-CT:

“Cancellous (trabecular) bone was assessed in the distal metaphysis and cortical bone was assessed in the mid-diaphysis. The femoral metaphysis region began 1,700 μm proximal to the distal growth plate and extended 1,500 μm distally.”

9. The age of the DKO mice (6-9 weeks) used in Figures 3 and 4 is not appropriate to compare with the YKL-05-099-treated OVX mice (24 weeks old). The DKO mice are young, rapidly growing mice, as they note on p.6, 2nd paragraph. But then, they refer to them as adult-onset DKO in the next paragraph, and in the title of the legends. This is not correct; the mice will have just reached puberty at 6 weeks when the inducible DKO occurs. They must repeat this study with mice the same age as the OVX mice. They may obtain a milder effect of SIK deletion on the bone phenotype.

In response to this very important comment, we have now performed additional experiments where *Sik2* and *Sik3* were deleted in older (12 week old) animals followed by analysis 4 weeks later. As shown in new Figure 3—figure supplement 3A-I, these mice show dramatic changes in trabecular bone and serum bone turnover markers that are largely consistent with what was observed in younger mutant animals. Based on the reviewer’s suggestion, the figure legends have been modified to indicate that we are studying postnatal-onset of global Sik2/3 deletion in our studies in Figures 3 and 4. Deletion of *Sik2/3* in 12 week old mice also did not change serum glucose or BUN levels, as indicated in the manuscript text.

10. Why did they examine the metabolic parameters at 8 weeks of age, not 9, which is when they killed the other mice? Also, were all the mice fasted for the same 3 h? What time is ZT3? Perhaps the increase in blood glucose and BUN caused by YKL-05-099 were because of the age differences?

Thank you for raising this point. We analyzed metabolic parameters at 2 weeks post tamoxifen, a time point when ubiquitin-Cre^ERt2^-mediated gene deletion is clearly noted. We reasoned that shorter time periods may be preferred for potential changes in serum parameters, while slightly longer time points may be better suited to phenotypic skeletal analyses. ZT3 means 3 hours after lights are turned on (~10am in our animal facility).

11. In Figure 5, it seems that the 4-OHT has effects on the abundance of SIK2 and SIK3, and there is less protein in vehicle-treated DKO cells, suggesting the Cre is leaky. They also need to show the abundance of SIK1.

Unfortunately, there are no commercially-available SIK1 antibodies that are suitable to detect endogenous murine SIK1 protein. For this reason, we developed a genomic DNA based assay to assess SIK1 gene deletion in vitro (Figure 5A, 6A) and in bone genomic DNA samples (Figure 3A). We agree that the immunoblotting data shown in Figure 5B suggested possible differences in protein loading. For this reason, we repeated this experiment, and in revised Figure 5C now show consistent loading and clear 4-OHT-mediated SIK2 and SIK3 protein deletion in DKO bone marrow macrophages.

12. Figure 7. shows that YKL-05-099 may inhibit a number of tyrosine kinases. They need to address the possibility that some of the effects they observe are due to other kinases than SIKs and the CSF1R. The effect on the growth plate may be similar to that of gefitinib and the off-target effects could be due to inhibition of other kinases.

We thank the reviewer for raising this important point. While changes in growth plate chondrocyte hypertrophy were observed in *Sik2/3^f/f^* ; ubiquitin-Cre^ERt2^ mice (Figure 3B, 3E, 4A, and S5), we did not observe changes in growth plate or bone length in mice treated with YKL-05-099 (Figure 2). Revised Supplemental Table 1 now shows comparable femur bone length measurements in all mice irrespective of drug treatment. We acknowledge the possibility that additional off-target effects may be responsible for the biologic activity seen with YKL-05-099, as discussed and addressed in more detail below (see response to Reviewer 3, point 2).

13. Why did they use such a high dose (15 uM) of YKL-05-099 in Figure 8E, F and G? This might be toxic.

This is an excellent point. Initially, we chose to use high doses of YKL-05-099 in these short-term studies in order to see clear treatment effects. In response to this well-taken criticism, we have performed additional experiments to establish the dose response relationship for the effects of YKL-05-099 on M-CSF signaling in bone marrow macrophages. In new Figure 8—figure supplement 1, we now show data indicating that sub-µM doses of YKL-05-099 also show similar inhibitory effects on M-CSF action in short term signaling (Figure 8—figure supplement 1A) and gene expression (Figure 8—figure supplement 1B) studies.

14. How many times were the in vitro experiments repeated. This needs to be mentioned in the methods or legends.

We thank the reviewer for raising this point. We have now added information in all relevant figure legends indicating the number of times each in vitro study was repeated.

Reviewer #2 (Recommendations for the authors):The authors should present osteoclast surface or number in figure 2D because ES/BS can be affected by changes in bone formation. Specifically, an increase in bone formation can reduce eroded surface even if osteoclast number does not change.

This is an excellent point. We have revised Figure 2 to now include histomorphometry data measuring osteoclast numbers (N.Oc/B.Pm) which previously was only shown in Supplemental Table 3. In general, osteoclast numbers correlate with eroded surface measurements in our histomorphometry results.

Reviewer #3 (Recommendations for the authors):In this study, Tang and colleague report that the multikinase inhibitor YKL-05-099 increases bone formation and decreases bone resorption in hypogonadal female mice with mechanisms that are likely to involve inhibition of SIKs and CSFR1, respectively. The authors also report that postnatal mice with inducible, global deletion of SIK2 and SIK3 show an increase of bone mass that is associated to both an augmentation of bone formation and bone resorption.The paper provides novel and interesting information with potentially highly relevant translational implications. The quality of the data is outstanding and most of the authors' conclusions are supported by the data as shown.1. Measurements of osteoclast number upon pharmacological treatments should be provided.

Figure 2 has been revised to include osteoclast numbers (these data were previously only shown in Supplemental Table 3), a request also made by the other two reviewers.

2. What is the experimental evidence that the increase of bone formation observed in mice treated with YKL-05-099 is mediated by SIKs inhibition? It would be helpful if the authors could discuss this point in the Discussion section.

This is an important point. Unfortunately, it is not possible to rigorously demonstrate that the bone anabolic effect of YKL-05-099 is due to targeting salt inducible kinases. The fifth paragraph of the Discussion has been revised accordingly:

“Finally, while both Sik2/3 deletion and YKL-05-099 treatment potently stimulate bone formation, we acknowledge that future study is needed to definitively prove that the bone anabolic effect of YKL-05-099 in vivo is due to targeting these two kinases. Currently, reagents (such as mice expressing an inhibitor-resistant SIK2^T96Q^ allele (63)) to rigorously address this question are not available.”